# Towards Understanding The Calibration Benefits of Sharpness-Aware Minimization

**Chengli Tan**[1,2]**, Yubo Zhou**[2,3]**, Haishan Ye**[2,3,*]**, Guang Dai**[2]**, Junmin Liu**[2,3,*]

**Zengjie Song**[3]**, Jiangshe Zhang**[3]**, Zixiang Zhao**[4]**, Yunda Hao**[5]**, Yong Xu**[1]

[1] Northwestern Polytechnical University    [2] SGIT AI Lab, State Grid Corporation of China
[3] Xi'an Jiaotong University    [4] ETH Zürich    [5] Chinese University of Hong Kong

## ABSTRACT

Deep neural networks have been increasingly used in safety-critical applications such as medical diagnosis and autonomous driving. However, many studies suggest that they are prone to being poorly calibrated and have a propensity for overconfidence, which may have disastrous consequences. In this paper, unlike standard training such as stochastic gradient descent, we show that the recently proposed sharpness-aware minimization (SAM) counteracts this tendency towards overconfidence. The theoretical analysis suggests that SAM allows us to learn models that are already well-calibrated by implicitly maximizing the entropy of the predictive distribution. Inspired by this finding, we further propose a variant of SAM, coined as CSAM, to ameliorate model calibration. Extensive experiments on various datasets, including ImageNet-1K, demonstrate the benefits of SAM in reducing calibration error. Meanwhile, CSAM performs even better than SAM and consistently achieves lower calibration error than other approaches.

## 1 INTRODUCTION

While the relation between generalization and flatness is still in dispute (Dinh et al., 2017; Ramasinghe et al., 2023; Andriushchenko et al., 2023; Wen et al., 2024), it is empirically appreciated that under some constraints, the flatter solutions tend to generalize better (Hinton & van Camp, 1993; Keskar et al., 2017; Chaudhari et al., 2019; Kaddour et al., 2022). From this point of view, many approaches have been proposed to bias solutions toward flat regions of the loss landscape explicitly or implicitly (Huang et al., 2017a; Izmailov et al., 2018; Chaudhari et al., 2019; Zhang et al., 2019; Wang et al., 2021b; Bisla et al., 2022), amongst which SAM (Foret et al., 2021) has garnered increasing attention due to its surprising effectiveness on popular tasks such as image classification (Chen et al., 2022), language generation (Bahri et al., 2022), and even physical computation (Xu et al., 2024).

Different from standard training like stochastic gradient descent (SGD), SAM minimizes a perturbed loss, and each iteration is composed of two consecutive steps,

$$\tilde{\boldsymbol{\theta}}_t = \boldsymbol{\theta}_t + \rho \frac{\nabla L_{\Omega_t}(\boldsymbol{\theta}_t)}{\|\nabla L_{\Omega_t}(\boldsymbol{\theta}_t)\|_2}, \quad \boldsymbol{\theta}_{t+1} = \boldsymbol{\theta}_t - \eta \nabla L_{\Omega_t}(\tilde{\boldsymbol{\theta}}_t),$$

where $\boldsymbol{\theta}_t \in \mathbb{R}^d$ represents the learnable parameters of the neural network at $t$-th iteration, $\eta$ is the learning rate, $\rho$ is the perturbation radius, and $L_{\Omega_t}(\cdot)$ denotes the empirical loss on a mini-batch $\Omega_t$ of the training set $S$. This scheme constantly penalizes the gradient norm (Zhao et al., 2022a; Wen et al., 2022; Compagnoni et al., 2023) and significantly promotes generalization. On the other hand, model calibration refers to how reliable the model predictions are. Ideally, when the model is confident about its predictions, the predictions are supposed to be as accurate as possible. This is particularly important for real-world applications such as autonomous driving (Chib & Singh, 2023) and medical diagnosis (Jiang et al., 2012). As an example, consider a self-driving car that uses deep neural networks to detect whether an obstruction is a pedestrian or not. For an ill-calibrated model,

---

*Corresponding authors (junminliu,yehaishan@mail.xjtu.edu.cn).

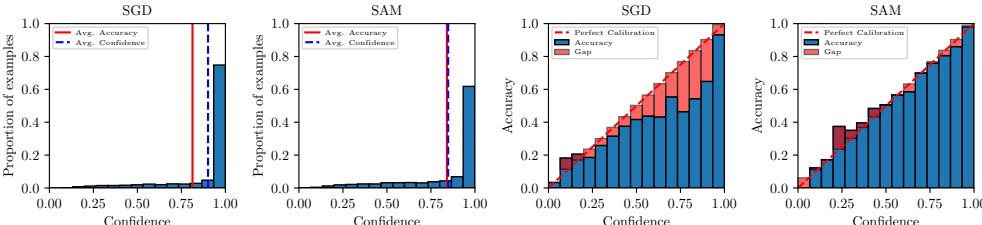

Figure 1: Confidence and reliability histograms for a PyramidNet (Han et al., 2017) trained on CIFAR-100 (Krizhevsky et al., 2009) with different optimizers. For clarity, the term *confidence* here refers to the predicted probability, namely, the maximum output of the softmax layer.

when its confidence is low, it may just pass through and will not trigger emergency braking, which could cause undesired consequences. In contrast, for a well-calibrated model, it is not certain whether the obstruction is a pedestrian or not when its confidence is low. As a result, a more cautious decision would be made by the car to avoid an accident.

It is known that modern neural networks such as ResNets (He et al., 2016) and DenseNets (Huang et al., 2017b) often suffer from the miscalibration problem, and this issue appears to be more serious when the network starts to overfit the training data (Nguyen et al., 2015; Guo et al., 2017; Zhu et al., 2023). Since SAM is more effective in preventing overfitting (Foret et al., 2021), one could anticipate that neural networks optimized by SAM may be better calibrated than by base optimizers such as SGD and AdamW (Loshchilov & Hutter, 2019). This is illustrated in Figure 1, where a large PyramidNet (Han et al., 2017) is respectively trained on CIFAR-100 (Krizhevsky et al., 2009) with SGD and SAM. One can easily observe that the average confidence of SAM closely matches its accuracy, while the average confidence of SGD is substantially higher than its accuracy. This is further confirmed with a reliability diagram (Niculescu-Mizil & Caruana, 2005), where we plot the accuracy as a function of the confidence. The diagram indicates that SAM is better calibrated than SGD, as the accuracy almost overlaps with the confidence along the diagonal line.

While previous studies (Zheng et al., 2021; Möllenhoff & Khan, 2023) have reported this phenomenon, the question of how SAM alleviates the miscalibration problem has not been formally investigated, and we attempt to fill this gap in this paper. In brief, our contributions are as follows:

- We provide theoretical justification for the calibration benefits of SAM, which essentially performs an implicit regularization on the negative entropy of the predictive distribution. This is similar to focal loss (Mukhoti et al., 2020), but SAM calibrates models much better without compromising accuracy.

- We investigate how SAM performs on model calibration under distribution shift and find that SAM allows models to remain well-calibrated under different types of corruption. Moreover, the trick of ensembling is also useful for SAM, and compared to SGD, the improvement is more pronounced on out-of-distribution data.

- We develop a variant of SAM, termed CSAM, that attempts to improve model calibration further. By extensive experiments with a variety of network architectures and datasets, we observe that CSAM consistently performs better than SAM and surpasses other approaches that are focused on improving calibration.

The remainder of the paper is organized as follows. We first review the related work in Section 2 and then introduce some backgrounds in Section 3. After presenting the theoretical analysis of SAM and the derivation of CSAM in Section 4, we further provide the experimental results in Section 5.

## 2 RELATED WORK

In this section, we present the most relevant works on SAM and the miscalibration of deep neural networks.

**Sharpness-aware minimization.** Because SAM is particularly effective in improving the generalization performance of realistic neural networks (Foret et al., 2021; Chen et al., 2022; Bahri et al., 2022), it has received a lot of attention in recent years, and there is a surge of research along this direction. For example, to reduce the computational overhead incurred by the additional backpropagation, some works choose to apply SAM and standard training alternatively (Liu et al., 2022b; Zhao et al., 2022b; Jiang et al., 2023; Tan et al., 2024a), while some other works focus on perturbing a fraction of parameters (Du et al., 2022; Mi et al., 2022) or examples (Ni et al., 2022). Concurrently, some researchers also attempt to further enhance the generalization performance of SAM (Zhang et al., 2022; Li & Giannakis, 2023; Yue et al., 2023; Zhou et al., 2023). For example, Kwon et al. (2021) propose ASAM to consolidate the correlation between sharpness and generalization, which might break up due to model reparameterization (Dinh et al., 2017). And Kim et al. (2022) further propose FisherSAM to enforce that the optimization occurs on the statistical manifold induced by the Fisher information matrix.

On the theoretical aspect, Wen et al. (2022); Bartlett et al. (2023) prove that the largest eigenvalue of the Hessian decreases along the trajectory of SAM, a result which is quite similar to that of Compagnoni et al. (2023) though derived from the perspective of the stochastic differential equation. Andriushchenko & Flammarion (2022) propose to study the unnormalized SAM and demonstrate the implicit bias on simple diagonal neural networks. Based on uniform stability (Bousquet & Elisseeff, 2002; Hardt et al., 2016), Tan et al. (2024b) prove that SAM generalizes better than SGD on strongly convex problems, and propose a renormalization trick to mitigate the instability issue near the saddle points (Compagnoni et al., 2023; Kim et al., 2023).

**Miscalibration of deep neural networks.** In machine learning, calibration has been extensively studied (Platt et al., 1999; Gneiting et al., 2007; Futami & Fujisawa, 2024). Since popular classification losses like squared error and cross-entropy (CE) are proper scoring rules (Gneiting et al., 2007), they are guaranteed to produce perfectly calibrated models at their global minimum. However, as first disclosed by Guo et al. (2017), modern neural networks suffer from serious miscalibration due to overfitting and overparameterization (Lakshminarayanan et al., 2017; Thulasidasan et al., 2019; Wang et al., 2021a; Wang, 2023). While Minderer et al. (2021) argue that the most recent non-convolutional models like MLP-Mixer (Tolstikhin et al., 2021) and vision transformers (Dosovitskiy et al., 2021) are better calibrated, the issue of miscalibration is still prevalent in a wide spectrum of applications like data distillation (Zhu et al., 2023) and object detection (Kuzucu et al., 2025).

A variety of approaches have been proposed to improve model calibration. In the training-time calibration, for example, an intuitive idea is to penalize overconfidence, either explicitly via entropy-based regularization (Pereyra et al., 2017) and label smoothing (Müller et al., 2019) or implicitly using focal loss (FL) (Mukhoti et al., 2020; Tao et al., 2023). However, as pointed out by previous works Wang et al. (2021a); Singh (2021), the penalty of confident outputs may suppress the potential improvement in the post-hoc calibration phase. On the other hand, post-hoc calibration addresses the miscalibration problem by appending a post-processing step to the training phase and typically requires a hold-out validation set for hyperparameter tuning. Popular post-hoc methods include non-parametric calibration methods—histogram binning (Zadrozny & Elkan, 2001) and isotonic regression (Zadrozny & Elkan, 2002), and parametric methods like Bayesian binning (Naeini et al., 2015) and Platt scaling (Platt et al., 1999). Out of them, Platt scaling-based approaches such as temperature scaling (Guo et al., 2017) and Dirichlet calibration (Kull et al., 2019) are more frequently used due to their low complexity and efficiency.

## 3 PRELIMINARIES

In this section, we first introduce one measure of model calibration that we use throughout, and then briefly recap the difference between SAM and SGD. Without loss of generality, we consider the multi-class classification problem where a categorical variable $Y \in \{1, \ldots, K\}$ is predicted when an input variable $X$ is observed. And we further assume that the training set $S$ contains $n$ examples $\{z_i = (x_i, y_i)\}_{i=1}^n$ that are *i.i.d.* sampled from an unknown data distribution $\mathcal{D}$. For a deep neural network parameterized by $\boldsymbol{\theta} \in \mathbb{R}^d$, we naturally obtain a predictor $f_{\boldsymbol{\theta}}$ that maps the features $X$ to a categorical distribution over $K$ labels, which we denote it by $f_{\boldsymbol{\theta}}(X)$ that belongs to a $(K-1)$-dimensional simplex $\Delta = \{\mathbf{p} \in [0,1]^K | \sum_{y=1}^K \mathbf{p}_y = 1\}$. Then, $\hat{y} \triangleq \arg\max_{1 \leq y \leq K} \mathbf{p}_y$ is the predicted label.

### 3.1 EXPECTED CALIBRATION ERROR

A model is well-calibrated if the confidence truthfully recovers the probability of correctness. That is, if we gather all data points for which the model predicts $\mathbf{p}_y = 0.8$, we expect that 80% of them should take on the label $y$. Mathematically, we refer to a model as well-calibrated (Bröcker, 2009) if

$$P(Y = y | f_{\boldsymbol{\theta}}(X) = \mathbf{p}) = \mathbf{p}_y, \quad \forall \, \mathbf{p} \in \Delta.$$

In practice, however, we will focus on the top-label calibration (Guo et al., 2017) that requires the above equation to hold only for the most likely label, namely,

$$P(Y = \hat{y} | \max_{1 \le y \le K} \mathbf{p}_y = \hat{p}) = \hat{p}, \quad \forall \, \hat{p} \in [0, 1].$$

Expected calibration error (ECE) is the most commonly used metric to measure the degree of miscalibration, which quantifies the expected difference between two sides of the above equation as follows

$$\mathbb{E} \left[ \left| \hat{p} - P(Y = \hat{y} | \max_{1 \le y \le K} \mathbf{p}_y = \hat{p}) \right| \right].$$

In practice, due to finite examples, it works by firstly grouping all examples, say, $\{z_i = (x_i, y_i)\}_{i=1}^n$, into $M$ bins $B_1, \ldots, B_M$ based on their top confidence scores. Next, we compute in each bin $B_i$ the average confidence $\mathrm{conf}(B_i) = 1/|B_i| \sum_{z_j \in B_i} \max f_{\boldsymbol{\theta}}(x_j)$ and the average accuracy $\mathrm{acc}(B_i) = 1/|B_i| \sum_{z_j \in B_i} \mathbb{I}[y_j = \arg\max f_{\boldsymbol{\theta}}(x_j)]$, where $\mathbb{I}[\cdot]$ is the indicator function. Then, we can obtain an estimator by averaging over the bins

$$\widehat{\mathrm{ECE}} = \sum_{i=1}^{M} \frac{|B_i|}{n} |\mathrm{acc}(B_i) - \mathrm{conf}(B_i)|.$$

### 3.2 SHARPNESS-AWARE MINIMIZATION

The intuitive idea of SAM (Foret et al., 2021) is to improve generalization by constantly minimizing the solution sharpness during training. To this end, instead of minimizing the loss at the current point, it minimizes the worst-case loss within its neighborhood. Mathematically, it is equivalent to solving the following optimization problem,

$$\min_{\boldsymbol{\theta} \in \mathbb{R}^d} \max_{\|\boldsymbol{\varepsilon}\|_2 \le \rho} L_S(\boldsymbol{\theta} + \boldsymbol{\varepsilon}),$$

where $\boldsymbol{\varepsilon} \in \mathbb{R}^d$ is a perturbation vector whose norm is bounded by the perturbation radius $\rho > 0$. It is not easy to solve this minimax problem explicitly. But, after a simple Taylor approximation, we observe that

$$\boldsymbol{\varepsilon}^* \triangleq \arg\max_{\|\boldsymbol{\varepsilon}\|_2 \le \rho} L_S(\boldsymbol{\theta} + \boldsymbol{\varepsilon})$$

$$\approx \arg\max_{\|\boldsymbol{\varepsilon}\|_2 \le \rho} L_S(\boldsymbol{\theta}) + \boldsymbol{\varepsilon}^T \nabla L_S(\boldsymbol{\theta}) = \rho \frac{\nabla L_S(\boldsymbol{\theta})}{\|\nabla L_S(\boldsymbol{\theta})\|_2}.$$

This suggests that, as opposed to SGD, we first need to do an extra gradient backpropagation to estimate the perturbed vector $\boldsymbol{\varepsilon}^*$. Therefore, SAM actually consists of two consecutive steps at each iteration,

$$\tilde{\boldsymbol{\theta}}_t = \boldsymbol{\theta}_t + \rho \frac{\nabla L_{\Omega_t}(\boldsymbol{\theta}_t)}{\|\nabla L_{\Omega_t}(\boldsymbol{\theta}_t)\|_2}, \quad \boldsymbol{\theta}_{t+1} = \boldsymbol{\theta}_t - \eta \nabla L_{\Omega_t}(\tilde{\boldsymbol{\theta}}_t),$$

where $\Omega_t$ denotes a random mini-batch of $S$. We note that the same $\Omega_t$ is used for the ascent and descent steps, and a smaller $\Omega_t$ is preferred in practice for better generalization (Foret et al., 2021; Andriushchenko & Flammarion, 2022).

## 4 METHODOLOGY

In this section, we first show that SAM is bound to prevent deep neural networks from producing overconfident predictions. As in previous studies (Guo et al., 2017; Minderer et al., 2021; Wang et al., 2021a), we focus on the most widely used cross-entropy (CE) loss in the classification problem, which for an example $z = (x, y)$ is defined as $\ell_{\boldsymbol{\theta}}(z) = -\log \mathbf{p}_y$ in one-hot encoding. The analysis is straightforward, and all proofs are deferred to Appendix A for clarity. Towards the end of this section, we also develop a variant of SAM to improve its calibration performance.

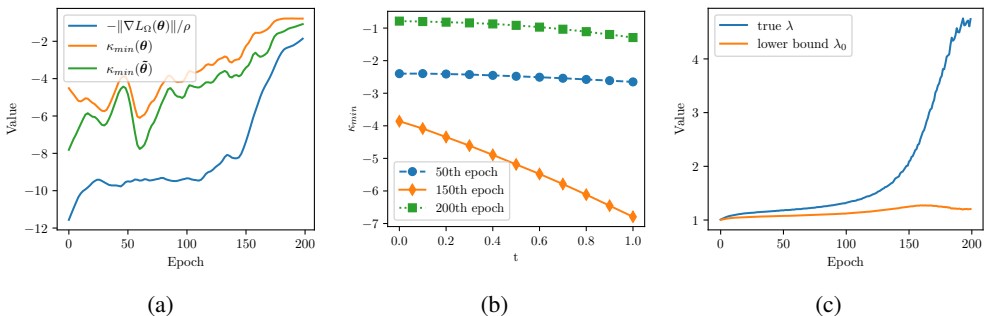

Figure 2: To verify whether the boundedness assumption of $\kappa_{min}$ (see Lemmas 1 and 2) holds for realistic neural networks, we trained a ResNet-56 on CIFAR-10 using a constant $\rho = 0.05$. (a) compares the value of $\kappa_{min}$ at the two endpoints $\boldsymbol{\theta}$ and $\tilde{\boldsymbol{\theta}}$ throughout the training process. (b) further illustrates how $\kappa_{min}$ evolves along the path from $\boldsymbol{\theta}(t = 0)$ to $\tilde{\boldsymbol{\theta}}(t = 1)$ at 50th, 150th, and 200th epoch, respectively. (c) records how the true coefficient $\lambda$ and its lower bound $\lambda_0$ (see Equation 2) vary during training. Notice that similar results for the realistic ImageNet-1K dataset can be found in Figure S4.

## 4.1 THEORETICAL ANALYSIS

Let $\mathbf{p}_y = [f_{\boldsymbol{\theta}}(x)]_y$ and $\tilde{\mathbf{p}}_y = [f_{\tilde{\boldsymbol{\theta}}}(x)]_y$ denote the confidence on the true label $y$ conditioned on the current weight $\boldsymbol{\theta}$ and the perturbed weight $\tilde{\boldsymbol{\theta}}$, respectively. When the mini-batch is 1, namely, every step we sample one example only to estimate the true gradient, the following lemma suggests that $\tilde{\mathbf{p}}_y$ can be consistently smaller than $\mathbf{p}_y$ during training.

**Lemma 1** (1-SAM version). *Assume that at each step, the gradient $\nabla_{\boldsymbol{\theta}}\ell(z) \neq \mathbf{0}$ and there always exists some $\rho > 0$ such that the smallest eigenvalue of the Hessian $\kappa_{min}(\nabla^2\ell_{\boldsymbol{\theta}'}(z)) \geq -\|\nabla_{\boldsymbol{\theta}}\ell(z)\|/\rho$ holds for all $\boldsymbol{\theta}' = (1 - t)\boldsymbol{\theta} + t\tilde{\boldsymbol{\theta}}$, $t \in [0, 1]$. Then, given $\mathbf{p}_y$, $\tilde{\mathbf{p}}_y$ defined as above, we have $\tilde{\mathbf{p}}_y \leq e^{-\rho\|\nabla_{\boldsymbol{\theta}}\ell(z)\|/2}\mathbf{p}_y$.*

Actually, the boundedness of $\kappa_{min}$ at $\boldsymbol{\theta}$ can be easily verified along the optimization trajectory (Zhou et al., 2021, Section 6.2). However, it should be noted that the inequality does not necessarily hold for all $\boldsymbol{\theta}'$. But if we vary $\rho$ accordingly at each step ($\rho \to 0$ in the worst case), the validity of the inequality can be assured because we are always ascending along the gradient direction. This lemma shows that $\tilde{\mathbf{p}}_y$, the probability of the perturbed network assigned to the true label, exponentially decays with the perturbation radius $\rho$ and the gradient norm $\|\nabla_{\boldsymbol{\theta}}\ell(z)\|$.

**Remark 1.** *A similar result for the mini-batch SAM is also developed in Lemma 2. Notice that varying $\rho$ at every step is quite different from the practical setting, in which we often use a constant $\rho$ instead. Luckily, as Figure 2(a) suggests, the boundedness assumption of $\kappa_{min}$ can be validated for the constant $\rho$ over mini-batch SAM. Surprisingly, we also find that $\kappa_{min}$ linearly decreases along $\boldsymbol{\theta}$ to $\tilde{\boldsymbol{\theta}}$ (see Figure 2(b)), suggesting that the boundedness assumption can be simplified to requiring $\kappa_{min}(\nabla^2\ell_{\tilde{\boldsymbol{\theta}}}(z)) \geq -\|\nabla_{\boldsymbol{\theta}}\ell(z)\|/\rho$ only. This finding further reveals why a large value of $\rho$ is not preferred because the boundedness assumption can be easily violated in that case.*

**Remark 2.** *It should be highlighted that the gradient norm $\|\nabla_{\boldsymbol{\theta}}\ell(z)\|$ also plays a critical role in determining $\tilde{\mathbf{p}}_y$. Lemma 1 indicates that the SAM optimizer is more effective for larger gradient norm, while simultaneously allowing us to choose a relatively large $\rho$. This finding is aligned with the observation that SAM is particularly effective in training ViT models with AdamW, which eventually improves more than 5% accuracy on ImageNet-1K using a large $\rho$ (Chen et al., 2022).*

Under Lemma 1, we show that minimizing the perturbed loss $\ell_{\tilde{\boldsymbol{\theta}}}(z)$ has the same effect as adding a maximum-entropy regularizer to $\ell_{\boldsymbol{\theta}}(z)$ as focal loss (FL) (Mukhoti et al., 2020, Section 4).

**Theorem 1** (1-SAM version). *Let $\lambda = (1 - \tilde{\mathbf{p}}_y)/(1 - \mathbf{p}_y)$, the following inequality holds*

$$\ell_{\tilde{\boldsymbol{\theta}}}(z) \geq \ell_{\boldsymbol{\theta}}(z) - \lambda H(\mathbf{p}_y) + H(\tilde{\mathbf{p}}_y), \tag{1}$$

*where $H(p) = -p\log p - (1 - p)\log(1 - p)$ is the binary entropy function.*

According to Lemma 1, we know that the coefficient $\lambda$ is larger than 1, which implies that minimizing $\ell_{\tilde{\theta}}(z)$ implicitly puts more emphasis on maximizing $H(\mathbf{p}_y)$ in contrast to minimizing $H(\tilde{\mathbf{p}}_y)$. That is, SAM forces $\mathbf{p}_y$ to be smaller when it approaches 1 and to be larger when it is near 0. Moreover, when replacing $\mathbf{p}_y$ with $e^{\rho\|\nabla_\theta \ell(z)\|/2}\tilde{\mathbf{p}}_y$, we have

$$\lambda \geq \lambda_0 = \frac{1 - \tilde{\mathbf{p}}_y}{1 - e^{\rho\|\nabla_\theta \ell(z)\|/2}\tilde{\mathbf{p}}_y}. \tag{2}$$

We note that the penalty on maximizing $H(\mathbf{p}_y)$ is stronger at the terminal phase of training than at the initial phase (see Figure 2(c)). Since model architecture is also a major determinant of model calibration (Minderer et al., 2021), it suggests that SAM could calibrate better for model architectures that are seriously overconfident.

In practice, as suggested by (Foret et al., 2021; Andriushchenko & Flammarion, 2022), we attempt to minimize the so-called $m$-sharpness to achieve the largest performance increment. Different from 1-SAM, in every step we determine the ascent direction using the gradient averaged over a mini-batch $\Omega$ of $m$ examples. As a result, the gradient corresponding to one example $\nabla\ell_\theta(z)$ is not promised to align well with the mini-batch gradient $\nabla L_\Omega(\theta) \triangleq 1/m \sum_{i=1}^m \nabla\ell_\theta(z_i)$. Therefore, the relation $\tilde{\mathbf{p}}_{y_i} \leq \mathbf{p}_{y_i}$ does not necessarily hold for all $z_i \in \Omega$. However, when both of them are taken into account, we do have a result similar to Lemma 1 as follows.

**Lemma 2** (m-SAM version). *Assume that at each step, the gradient $\nabla L_\Omega(\theta) \neq \mathbf{0}$ and there always exists some $\rho > 0$ such that the smallest eigenvalue of the Hessian $\kappa_{min}(\nabla^2 L_\Omega(\theta')) \geq -\|\nabla L_\Omega(\theta)\|/\rho$ holds for all $\theta' = (1-t)\theta + t\tilde{\theta}$, $t \in [0,1]$. Denote $\mathbf{p}_y = (\prod_{i=1}^m \mathbf{p}_{y_i})^{1/m}$ and $\tilde{\mathbf{p}}_y = (\prod_{i=1}^m \tilde{\mathbf{p}}_{y_i})^{1/m}$, respectively. Then, we have $\tilde{\mathbf{p}}_y \leq e^{-\rho\|L_\Omega(\theta)\|/2}\mathbf{p}_y$.*

The proof is straightforward, and accordingly, we have the following result.

**Theorem 2** (m-SAM version). *Let $\mathbf{p}_y$ and $\tilde{\mathbf{p}}_y$ defined as above. Then, it follows that*

$$L_\Omega(\tilde{\theta}) \geq L_\Omega(\theta) - \lambda H(\mathbf{p}_y) + H(\tilde{\mathbf{p}}_y), \tag{3}$$

*where $\lambda = (1 - \tilde{\mathbf{p}}_y)/(1 - \mathbf{p}_y)$.*

This theorem is similar to Theorem 1, albeit $\mathbf{p}_y$ is the geometric mean of the predicted probabilities. But it is enough to make sure that $m$-SAM prevents models from producing overconfident predictions as well.

## 4.2 IMPROVING SAM TOWARDS BETTER CALIBRATION

As shown in Figure 2(c), we notice that SAM primarily starts to penalize the predictive distribution at the late stages of training where $\tilde{\mathbf{p}}_y$ is high. Therefore, we propose to suppress the contribution of the over-confident examples so that their predictive probability $\tilde{\mathbf{p}}_y$ is virtually higher. That is, we can redefine the per-example loss function for the outer loop of SAM as follows:

$$\tilde{\ell}_{\tilde{\theta}}(z) = \begin{cases} -\log \tilde{\mathbf{p}}_y, & \text{if } \tilde{\mathbf{p}}_y \leq 1/2, \\ -(1 + \tilde{\mathbf{p}}_y)^{-\gamma} \log \tilde{\mathbf{p}}_y, & \text{otherwise,} \end{cases} \tag{4}$$

where $0 \leq \gamma \leq 2$ is a hyperparameter. It is trivial to recover the standard SAM when $\gamma = 0$. Actually, the following result suggests that the modified loss function $\tilde{\ell}_{\tilde{\theta}}(z)$ enforces SAM to penalize the predictive distribution of over-confident examples.

**Theorem 3.** *Let Lemma 1 hold and $\lambda = (1 - \tilde{\mathbf{p}}_y)/(1 - \mathbf{p}_y)$, for all $\tilde{\mathbf{p}}_y > 1/2$, the following inequality holds*

$$\tilde{\ell}_{\tilde{\theta}}(z) \geq \ell_\theta(z) - \lambda H(\mathbf{p}_y) + (1 - \gamma/2)H(\tilde{\mathbf{p}}_y), \tag{5}$$

*where $H(p) = -p \log p - (1-p)\log(1-p)$ is the binary entropy function.*

Slightly different from Theorem 1, here it brings a coefficient before $H(\tilde{\mathbf{p}}_y)$, which suggests that the implicit penalty on $H(\mathbf{p}_y)$ is stronger if $(1 - \gamma/2) > 0$. Meanwhile, we also require that $\gamma \leq 2$ so that the optimization process is always biased towards decreasing $\ell_{\tilde{\theta}}(z)$ as in SAM. Note that this argument is also valid for $m$-SAM as it increases the geometric mean as well. For notational convenience, we will refer to this variant of SAM as Calibrated SAM (CSAM) in the sequel, and its pseudocode is summarized in Algorithm 1 (see Appendix A).

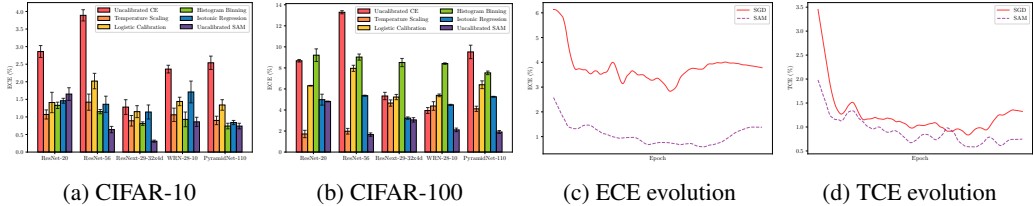

| (a) CIFAR-10 | (b) CIFAR-100 | (c) ECE evolution | (d) TCE evolution |

Figure 3: (a)-(b) display the calibration performance of SAM and SGD (after various post-hoc processing) on CIFAR-10/100 datasets. (c)-(d) report the variation of ECE and TCE (namely, ECE after temperature scaling) during training.

## 5 EXPERIMENTS

In this section, we present the experimental results. We begin with the standard benchmarks showing that SAM significantly calibrates better than SGD. We further demonstrate on datasets including ImageNet-1K (Deng et al., 2009) that this calibration benefit is not limited to the in-distribution (ID) data, but also translates to the out-of-distribution (OOD) data. At last, we compare the proposed CSAM and SAM against a variety of baselines that attempt to reduce miscalibration. The results suggest that SAM is competitive and even superior to these approaches in many cases. More surprisingly, our proposed CSAM consistently outperforms SAM and achieves the lowest calibration error out of all baselines without deteriorating the generalization performance.

### 5.1 SAM ATTAINS A LOWER CALIBRATION ERROR THAN SGD

As a starting point, we first evaluate how SAM differs from SGD on the classical benchmarks for classification. The loss function defaults to be the standard cross-entropy (CE) loss, and we train several neural networks, including ResNets (He et al., 2016), Wide ResNets (Zagoruyko & Komodakis, 2016), and PyramidNets (Han et al., 2017) to classify CIFAR-10/100 (Krizhevsky et al., 2009). As in common practice, we split the data into the train, validation, and test subsets so that the same validation subset is used for hyperparameter tuning and post-hoc calibration. Without further specification, the optimizer is SGD with momentum 0.9, and the learning rate is scheduled in a cosine decay (Loshchilov & Hutter, 2017). To conduct a fair comparison, we first make a grid search of learning rate and weight decay coefficient on the model trained with SGD, and then apply them to SAM. The perturbation radius $\rho$ is 0.05 for CIFAR-10 and 0.2 for CIFAR-100 (see Appendix B for more discussion on the effect of $\rho$ on calibration).

As illustrated in Figure 3(a)-(b), the ECE of SGD (red bar) is always much higher than the ECE of SAM (purple bar). This is more pronounced for ResNet-56 on CIFAR-10/100, where the ECE of SGD is approximately six times larger than the ECE of SAM. More surprisingly, we further observe that the uncalibrated ECE of SAM is generally smaller than the calibrated ECE of SGD by calibration methods such as temperature scaling (Guo et al., 2017) and isotonic regression (Zadrozny & Elkan, 2002). This indicates that SAM by itself tends to generate accurate and reliable predictions. Furthermore, as shown in Figure 3(c)-(d), the superiority of SAM is persistent across the full training process. And the reduction of ECE is more pronounced than TCE since SAM has already suppressed the over-confident outputs during training, and temperature scaling is thus not as effective as in SGD.

### 5.2 MODEL CALIBRATION UNDER DISTRIBUTION SHIFT

It is important for safety-critical applications that the model not only produces reliable predictions for the in-distribution data but also is robust enough when there exists a distribution shift between the training data and the test data. For this purpose, we first train ResNet-18 on CIFAR-10 using vanilla SGD and SAM, and then evaluate its performance on other datasets, including SVHN (Netzer et al., 2011), CIFAR-10/100-C (Hendrycks & Dietterich, 2019). To enhance model uncertainty, we further encapsulate them with MC-Dropout (Gal & Ghahramani, 2016) and Ensemble (Ovadia et al., 2019). Table 1 shows that model ensembling and MC-Dropout both can reduce ECE for SGD, SAM, and CSAM, but their gap is still significant—ECE of SGD approximately remains two times larger than ECE of SAM. This is different from their behavior on test accuracy, for example, SGD almost

Table 1: Model performance on OOD data. The base model is ResNet-18 trained on CIFAR-10. The size of MC-Dropout and Ensemble is 5.

| | | ID Metrics | | OOD AUROC ↑ | | |
|---|---|---|---|---|---|---|
| | | Test Acc ↑ | ECE ↓ | SVHN | CIFAR10-C | CIFAR100-C |
| SGD | Vanilla | $89.18 \pm 0.26$ | $5.76 \pm 0.43$ | $83.94 \pm 0.96$ | $62.26 \pm 4.46$ | $83.26 \pm 0.71$ |
| | MC-Dropout | $89.13 \pm 0.18$ | $4.39 \pm 0.27$ | $84.11 \pm 0.69$ | $57.09 \pm 2.81$ | $82.11 \pm 0.82$ |
| | Ensemble | $90.88 \pm 0.11$ | $1.84 \pm 0.22$ | $86.41 \pm 0.36$ | $63.39 \pm 4.72$ | $85.81 \pm 0.17$ |
| SAM | Vanilla | $90.01 \pm 0.23$ | $3.24 \pm 0.39$ | $86.38 \pm 0.39$ | $63.32 \pm 4.77$ | $84.83 \pm 0.83$ |
| | MC-Dropout | $89.49 \pm 0.33$ | $2.21 \pm 0.41$ | $83.02 \pm 0.35$ | $56.11 \pm 2.50$ | $80.96 \pm 0.55$ |
| | Ensemble | $91.16 \pm 0.14$ | $1.09 \pm 0.22$ | $88.05 \pm 0.21$ | $64.03 \pm 4.95$ | $86.84 \pm 0.75$ |
| CSAM | Vanilla | $89.95 \pm 0.16$ | $2.55 \pm 0.24$ | $85.98 \pm 0.42$ | $63.49 \pm 4.74$ | $84.87 \pm 0.86$ |
| | MC-Dropout | $89.57 \pm 0.21$ | $1.52 \pm 0.21$ | $82.82 \pm 0.31$ | $56.05 \pm 2.48$ | $80.70 \pm 0.55$ |
| | Ensemble | $\mathbf{91.22 \pm 0.17}$ | $\mathbf{0.86 \pm 0.17}$ | $\mathbf{88.21 \pm 0.12}$ | $\mathbf{64.17 \pm 4.91}$ | $\mathbf{86.92 \pm 0.70}$ |

Table 2: Results on the ImageNet-1K dataset. Slightly different from the custom setting, we reserve 20% of the ImageNet-1K validation set as a new validation set for early stopping and temperature scaling, and the remaining images therefore constitute a test set. Both metrics (TCE is short for ECE calibrated by temperature scaling, and AdaECE is adaptive ECE) are evaluated on the test set.

| | | ID Metrics | | | | OOD Metrics | | | | | |
|---|---|---|---|---|---|---|---|---|---|---|---|
| | | Test Acc ↑ | ECE ↓ | TCE ↓ | AdaECE ↓ | AUROC ↑ | Test Acc (1/2/3) ↑ | | | ECE (1/2/3) ↓ | | |
| ResNet-50 | SGD | 76.97 | 3.39 | 1.80 | 3.31 | 94.01 | 36.89 | 35.81 | 24.99 | 7.97 | 4.23 | 17.29 |
| | SAM | 77.32 | 1.52 | 1.54 | 1.44 | 94.35 | 37.45 | 36.35 | 27.85 | 4.91 | 3.74 | 6.92 |
| | CSAM | **77.95** | **1.18** | **1.09** | **1.19** | **94.67** | **38.29** | **37.11** | **28.69** | **3.28** | **3.02** | **5.47** |
| ViT-S/32 | AdamW | 65.03 | 9.11 | 2.63 | 9.11 | 88.63 | 33.53 | 32.87 | 26.48 | 14.73 | 12.28 | 19.57 |
| | SAM | 69.21 | 3.04 | 1.18 | 3.05 | 91.01 | 37.95 | 36.09 | 33.36 | 3.35 | 6.27 | 7.89 |
| | CSAM | **70.01** | **2.88** | **0.92** | **2.78** | **91.54** | **38.88** | **36.94** | **34.16** | **3.01** | **5.76** | **5.41** |
| ViT-S/16 | AdamW | 71.35 | 9.72 | 3.66 | 9.72 | 90.61 | 37.40 | 35.54 | 24.26 | 14.14 | 12.27 | 18.63 |
| | SAM | 75.42 | 1.76 | 1.66 | 1.73 | 93.27 | 43.36 | 39.15 | 28.93 | 2.92 | 3.58 | 5.02 |
| | CSAM | **75.91** | **1.58** | **1.34** | **1.54** | **93.66** | **44.01** | **39.82** | **29.57** | **2.81** | **3.24** | **4.75** |

generalizes as well as SAM with Ensemble. On the other hand, it should be highlighted that SAM generalizes much better than SGD on OOD data. And Ensemble also works well under this scenario. An unexpected finding is that MC-Dropout hurts both optimizers' performance on OOD data and is more evident for SAM. One possible explanation is that the fusion of Dropout and SAM adversely increases model uncertainty, which, as a result, impedes generalization.

Next, we train models on the clean ImageNet-1K dataset and then assess the calibration performance of SAM on the ImageNet-C (Hendrycks & Dietterich, 2019) dataset, which consists of images that have been modified with several synthetic corruptions at five different severities. Following Minderer et al. (2021), we reserve 20% of the ImageNet-1K validation set for early stopping and temperature scaling. Moreover, we also exclude the corresponding corrupted images in ImageNet-C that are created from ImageNet-1K at the evaluation phase. We train one ResNet and two vision transformers (ViTs) (Dosovitskiy et al., 2021) on ImageNet-1K for 100 epochs and 300 epochs. The base optimizers are SGD and AdamW, and a cosine learning rate scheduler is used in all runs. As in previous studies (Foret et al., 2021; Chen et al., 2022), the perturbation radius $\rho$ for ResNet and ViT is 0.05 and 0.2.

As shown in Table 2, SAM and CSAM consistently improve the test accuracy on ImageNet-1K validation set, though being more pronounced for ViTs ($\sim 4\%$). Meanwhile, ViTs are generally less calibrated than ResNet, which is somewhat inconsistent with the findings of (Minderer et al., 2021). One explanation might be that their comparison is based on the pretrained neural networks rather than training them from scratch. But when models are trained by SAM, both of them achieve a much lower calibration error, and their gap becomes negligible. For ImageNet-C, we consider three kinds of corruption: 1–motion blur, 2–defocus blur, and 3–impulse noise. For each kind of corruption, we further average the accuracy and ECE across the five different severities. Consistent with previous findings, Table 2 also indicates that SAM generalizes better than SGD and that ViTs trained by AdamW also tend to be less calibrated on ImageNet-C. Interestingly, however, we observe

Table 3: Performance comparison between different methods on CIFAR-10. The results are averaged over 3 random seeds, with standard deviation displayed as well.

|  | Test Acc ↑ | ECE ↓ | ClasswiseECE ↓ | AdaECE ↓ | TCE ↓ | AUROC ↑ |
|---|---|---|---|---|---|---|
| CE | $95.83 \pm 0.21$ | $2.36 \pm 0.11$ | $0.52 \pm 0.01$ | $2.04 \pm 0.11$ | $1.06 \pm 0.19$ | $98.68 \pm 0.04$ |
| Focal Loss (FL) | $95.91 \pm 0.02$ | $1.16 \pm 0.13$ | $0.38 \pm 0.01$ | $1.42 \pm 0.09$ | $1.01 \pm 0.28$ | $99.04 \pm 0.01$ |
| DualFocal | $95.73 \pm 0.10$ | $1.74 \pm 0.09$ | $0.48 \pm 0.02$ | $1.64 \pm 0.07$ | $1.00 \pm 0.09$ | $99.26 \pm 0.02$ |
| AdaFocal | $95.78 \pm 0.06$ | $0.91 \pm 0.14$ | $0.35 \pm 0.01$ | $0.65 \pm 0.04$ | $0.97 \pm 0.08$ | $99.10 \pm 0.04$ |
| Mixup | $96.34 \pm 0.10$ | $2.21 \pm 1.11$ | $0.45 \pm 0.21$ | $1.63 \pm 1.04$ | $1.33 \pm 0.25$ | $99.12 \pm 0.02$ |
| MIT-L | $96.56 \pm 0.16$ | $1.05 \pm 0.02$ | $0.31 \pm 0.01$ | $1.05 \pm 0.05$ | $0.57 \pm 0.11$ | $99.12 \pm 0.03$ |
| MMCE | $95.94 \pm 0.02$ | $2.47 \pm 0.04$ | $0.54 \pm 0.02$ | $2.42 \pm 0.04$ | $1.15 \pm 0.18$ | $98.65 \pm 0.05$ |
| BatchEnsemble | $95.92 \pm 0.11$ | $1.91 \pm 0.06$ | $0.45 \pm 0.01$ | $1.85 \pm 0.03$ | $0.41 \pm 0.01$ | $98.96 \pm 0.01$ |
| Rank1-BNN | $95.50 \pm 0.14$ | $1.92 \pm 0.29$ | $0.45 \pm 0.06$ | $1.94 \pm 0.29$ | $0.51 \pm 0.03$ | $98.81 \pm 0.12$ |
| VI | $94.33 \pm 0.10$ | $3.14 \pm 0.12$ | $0.69 \pm 0.03$ | $3.06 \pm 0.15$ | $0.76 \pm 0.08$ | $98.28 \pm 0.07$ |
| MIMO | $95.96 \pm 0.06$ | $0.88 \pm 0.06$ | $0.33 \pm 0.01$ | $0.73 \pm 0.08$ | $0.74 \pm 0.20$ | $99.16 \pm 0.01$ |
| ACLS | $95.91 \pm 0.08$ | $2.48 \pm 0.07$ | $0.55 \pm 0.01$ | $2.45 \pm 0.07$ | $1.09 \pm 0.08$ | $98.62 \pm 0.01$ |
| BalCAL | $96.23 \pm 0.09$ | $1.89 \pm 0.07$ | $0.42 \pm 0.01$ | $1.93 \pm 0.01$ | $0.79 \pm 0.05$ | $98.77 \pm 0.44$ |
| bSAM | $96.45 \pm 0.03$ | $1.82 \pm 0.10$ | $0.43 \pm 0.02$ | $1.78 \pm 0.10$ | $0.70 \pm 0.23$ | $98.95 \pm 0.06$ |
| SAM | $96.91 \pm 0.14$ | $0.86 \pm 0.13$ | $0.26 \pm 0.02$ | $0.84 \pm 0.14$ | $0.52 \pm 0.09$ | $99.30 \pm 0.02$ |
| CSAM | $\mathbf{96.97 \pm 0.05}$ | $\mathbf{0.50 \pm 0.03}$ | $\mathbf{0.23 \pm 0.01}$ | $\mathbf{0.48 \pm 0.03}$ | $\mathbf{0.47 \pm 0.05}$ | $\mathbf{99.53 \pm 0.02}$ |

that while ViT-S/16-SAM generalizes and calibrates worse than ResNet-50-SAM, it performs much better than the latter. This might arise from the different implicit biases of SGD and AdamW.

### 5.3 CSAM EVEN CALIBRATES BETTER THAN SAM

In this section, we attempt to compare CSAM and SAM against other popular baselines, including: focal loss (Mukhoti et al., 2020) that implicitly penalizes the gradient norms of confident examples and its two variants—DualFocal (Tao et al., 2023) and AdaFocal (Ghosh et al., 2022), mixup (Zhang, 2018) that implicitly performs label smoothing (Carratino et al., 2022) to avoid the overconfidence issue, MMCE (Kumar et al., 2018) that acts as a continuous and differentiable calibration error regulariser, MIT-L (Wang et al., 2023) that involves mixup inference in training, BatchEnsemble (Wen et al., 2020), ACLS (Park et al., 2023), BalCAL (Ni et al., 2025), and several probabilistic approaches—Rank1-BNN (Dusenberry et al., 2020), VI (Ovadia et al., 2019), MIMO (Havasi et al., 2021), and bSAM (Möllenhoff & Khan, 2023). The backbone is WideResNet-20-10 (Zagoruyko & Komodakis, 2016), and we generally follow the recommended setting to reproduce the results of each baseline. The perturbation radius $\rho$ of SAM and CSAM is 0.2 for CIFAR-10/100, and we vary the hyper-parameter $\gamma$ of CSAM in $\{0.5, 1.0, 2.0\}$.

From Tables 3 and S5, we can observe that while focal loss generally hurts generalization, it does reduce the calibration error. This observation also applies to the probabilistic approaches, such as Rank1-BNN and MIMO. As a comparison, SAM significantly reduces the calibration error and is competitive, even superior to other baselines in many cases. Note that the Bayesian variant, bSAM, does not perform better than SAM. The reason might be that it additionally introduces several hyperparameters, making it more difficult to tune and apply. In contrast, the proposed CSAM further decreases the calibration error while simultaneously achieving a competitive generalization performance to SAM. And when compared to other baselines, CSAM always achieves the lowest error, showing its versatility in generalization and calibration. While our current study is limited to the cross-entropy loss, preliminary studies (see Table S15) indicate that SAM/CSAM can be further integrated with other training losses. More results, such as sensitivity analysis of the hyperparameters and comparison to other variants of SAM, can be found in Appendices B and C.

## 6 CONCLUSION

Besides its well-known generalization benefits, we showed that SAM also excels at calibrating deep neural networks. We proved that SAM achieves this goal by imposing an implicit regularization on the negative entropy of the predictive distribution during training (see Equation 1), which is similar to focal loss (Mukhoti et al., 2020). We further proposed a variant of SAM to improve calibration and validated its superiority across a number of networks and datasets.

## ACKNOWLEDGMENTS

This work was supported in part by the National Natural Science Foundation of China under Grants 12501710, 12301656, 62276208, 12326607 and 12371512, in part by the Natural Science Basic Research Program of Shaanxi Province under Grant 2024JC-JCQN-02, and in part by Fundamental Research Funds for the Central Universities.

## DISCLOSURE OF THE USE OF LARGE LANGUAGE MODELS

The authors use large language models to assist with copyediting and to improve the overall readability of the text. The authors review and edit the manuscript and assume full responsibility for its final content.

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

---

**Algorithm 1** CSAM Optimizer

---

**Input:** Training set $S = \{z_i = (x_i, y_i)\}_{i=1}^n$, objective function $L_S(\boldsymbol{\theta})$, initial weight $\boldsymbol{\theta}_0 \in \mathbb{R}^d$, learning rate $\eta > 0$, perturbation radius $\rho > 0$, training iterations $T$, regularization coefficient $\gamma > 0$, and base optimizer $\mathcal{A}$ (e.g. SGD)

**Output:** $\boldsymbol{\theta}_T$

1: **for** $t = 0, 1, \cdots, T - 1$ **do**
2:      Sample a mini-batch $\Omega_t = \{z_1^t, \cdots, z_m^t\}$;
3:      Compute cross-entropy loss $L_{\Omega_t}(\boldsymbol{\theta_t}) = \frac{1}{m} \sum_{z_i \in \Omega_t} \ell_{\tilde{\boldsymbol{\theta}}}(z_i)$;
4:      Compute perturbed weight $\tilde{\boldsymbol{\theta}}_t = \boldsymbol{\theta}_t + \rho \cdot \frac{\nabla_{\boldsymbol{\theta}_t} L_{\Omega_t}(\boldsymbol{\theta}_t)}{\|\nabla_{\boldsymbol{\theta}_t} L_{\Omega_t}(\boldsymbol{\theta}_t)\|}$;
5:      Compute perturbed loss $L_{\Omega_t}(\tilde{\boldsymbol{\theta}}_t) = \frac{1}{m} \sum_{z_i \in \Omega_t} \tilde{\ell}_{\tilde{\boldsymbol{\theta}}}(z_i)$ per Equation (4);
6:      Compute gradient $\tilde{\boldsymbol{g}}_t = \nabla_{\boldsymbol{\theta}} L_{\Omega_t}(\tilde{\boldsymbol{\theta}}_t)|_{\boldsymbol{\theta} = \tilde{\boldsymbol{\theta}}_t}$ of the loss over the same $\Omega_t$;
7:      Update weight with base optimizer $\mathcal{A}$, e.g. $\boldsymbol{\theta}_{t+1} = \boldsymbol{\theta}_t - \eta \tilde{\boldsymbol{g}}_t$;
8: **end for**

---

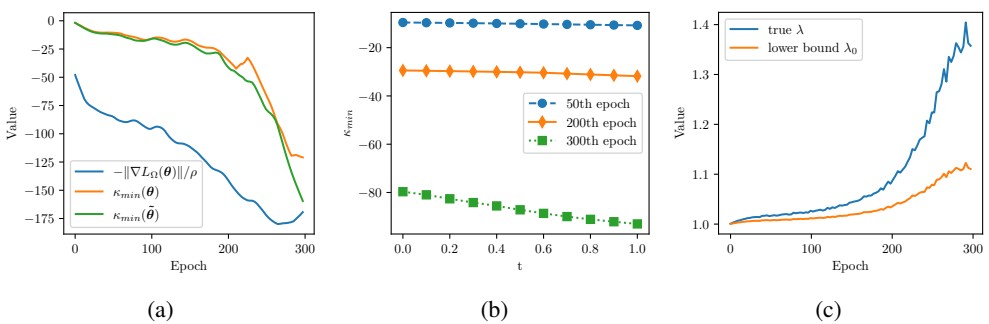

(a)            (b)            (c)

Figure S4: To verify whether the boundedness assumption of $\kappa_{min}$ (see Lemmas 1 and 2) holds for realistic neural networks, we further trained a ViT on the realistic ImageNet-1K dataset. (a) compares the value of $\kappa_{min}$ at the two endpoints $\boldsymbol{\theta}$ and $\tilde{\boldsymbol{\theta}}$ throughout the training process. (b) further illustrates how $\kappa_{min}$ evolves along the path from $\boldsymbol{\theta}(t = 0)$ to $\tilde{\boldsymbol{\theta}}(t = 1)$ at 50th, 200th, and 300th epoch, respectively. (c) records how the true coefficient $\lambda$ and its lower bound $\lambda_0$ (see Equation 2) vary during training.

## A    THEORETICAL PROOFS

In this section, we first present the pseudocode of CSAM (Algorithm 1) and then the missing proofs in Section 4. We also validate the boundedness assumption on the ImageNet-1K dataset, as shown in Figure S4.

**Proof of Lemma 1.** According to the Taylor theorem, there always exists some $\boldsymbol{\theta}'$ such that

$$-\log \tilde{\mathbf{p}}_y = \ell_{\tilde{\boldsymbol{\theta}}}(z) = \ell_{\boldsymbol{\theta}}(z) + (\tilde{\boldsymbol{\theta}} - \boldsymbol{\theta})^T \nabla \ell_{\boldsymbol{\theta}}(z) + \frac{1}{2}(\tilde{\boldsymbol{\theta}} - \boldsymbol{\theta})^T \nabla^2 \ell_{\boldsymbol{\theta}'}(z)(\tilde{\boldsymbol{\theta}} - \boldsymbol{\theta}).$$

Since $\tilde{\boldsymbol{\theta}} = \boldsymbol{\theta} + \rho \nabla \ell_{\boldsymbol{\theta}}(z)/\|\nabla \ell_{\boldsymbol{\theta}}(z)\|_2$ and $\kappa_{min}(\nabla^2 \ell_{\boldsymbol{\theta}'}(z)) > -\|\nabla \ell_{\boldsymbol{\theta}}(z)\|_2/\rho$, it follows that

$$-\log \tilde{\mathbf{p}}_y = \ell_{\tilde{\boldsymbol{\theta}}}(z) \geq \ell_{\boldsymbol{\theta}}(z) + \rho \|\nabla \ell_{\boldsymbol{\theta}}(z)\|_2 + \frac{\rho^2}{2} \kappa_{min}(\nabla^2 \ell_{\boldsymbol{\theta}'}(z)) \geq -\log \mathbf{p}_y + \frac{\rho}{2} \|\nabla \ell_{\boldsymbol{\theta}}(z)\|_2,$$

thus concluding the proof.

**Proof of Theorem 1.** It follows from Lemma 1 that $\tilde{\mathbf{p}}_y \leq \mathbf{p}_y$. Recall that

$$\ell_{\tilde{\boldsymbol{\theta}}}(z) = -\log \tilde{\mathbf{p}}_y = \ell_{\boldsymbol{\theta}}(z) + \log \frac{\mathbf{p}_y}{\tilde{\mathbf{p}}_y} \geq \ell_{\boldsymbol{\theta}}(z) + \tilde{\mathbf{p}}_y \log \frac{\mathbf{p}_y}{\tilde{\mathbf{p}}_y} + (1 - \tilde{\mathbf{p}}_y) \log \frac{1 - \mathbf{p}_y}{1 - \tilde{\mathbf{p}}_y}$$

$$\geq \ell_{\boldsymbol{\theta}}(z) - \frac{1 - \tilde{\mathbf{p}}_y}{1 - \mathbf{p}_y} H(\mathbf{p}_y) + H(\tilde{\mathbf{p}}_y),$$

thus concluding the proof.

**Proof of Lemma 2.** There always exists some $\boldsymbol{\theta}' \in \mathbb{R}^d$ such that

$$-\log\left(\prod_{i=1}^m \tilde{\mathbf{p}}_{y_i}\right)^{1/m} = -\frac{1}{m}\sum_{i=1}^m \log \tilde{\mathbf{p}}_{y_i}$$
$$= L_\Omega(\tilde{\boldsymbol{\theta}})$$
$$= L_\Omega(\boldsymbol{\theta}) + \left(\tilde{\boldsymbol{\theta}} - \boldsymbol{\theta}\right)^T \nabla L_\Omega(\boldsymbol{\theta}) + \frac{1}{2}\left(\tilde{\boldsymbol{\theta}} - \boldsymbol{\theta}\right)^T \nabla^2 L_\Omega(\boldsymbol{\theta}')\left(\tilde{\boldsymbol{\theta}} - \boldsymbol{\theta}\right).$$

A similar argument as Lemma 1 concludes the proof.

**Proof of Theorem 2.** The proof is straightforward. According to the definition of $\mathbf{p}_y$ and $\tilde{\mathbf{p}}_y$, it yields that

$$L_\Omega(\tilde{\boldsymbol{\theta}}) = -\log \tilde{\mathbf{p}}_y = L_\Omega(\boldsymbol{\theta}) + \log \frac{\mathbf{p}_y}{\tilde{\mathbf{p}}_y} \geq L_\Omega(\boldsymbol{\theta}) + \tilde{\mathbf{p}}_y \log \frac{\mathbf{p}_y}{\tilde{\mathbf{p}}_y} + (1 - \tilde{\mathbf{p}}_y)\log\frac{1 - \mathbf{p}_y}{1 - \tilde{\mathbf{p}}_y}$$
$$\geq L_\Omega(\boldsymbol{\theta}_k) - \frac{1 - \tilde{\mathbf{p}}_y}{1 - \mathbf{p}_y} H(\mathbf{p}_y) + H(\tilde{\mathbf{p}}_y),$$

thus completing the proof.

**Proof of Theorem 3.** Recall that

$$\tilde{\ell}_{\tilde{\boldsymbol{\theta}}}(z) = -(1 + \tilde{\mathbf{p}}_y)^{-\gamma}\log \tilde{\mathbf{p}}_y \geq -(1 - \gamma\tilde{\mathbf{p}}_y)\log \tilde{\mathbf{p}}_y$$
$$= \ell_{\tilde{\boldsymbol{\theta}}}(z) + \gamma\tilde{\mathbf{p}}_y \log \tilde{\mathbf{p}}_y$$
$$\geq \ell_{\boldsymbol{\theta}}(z) - \frac{1 - \tilde{\mathbf{p}}_y}{1 - \mathbf{p}_y}H(\mathbf{p}_y) + H(\tilde{\mathbf{p}}_y) + \gamma\tilde{\mathbf{p}}_y \log \tilde{\mathbf{p}}_y$$
$$\geq \ell_{\boldsymbol{\theta}}(z) - \frac{1 - \tilde{\mathbf{p}}_y}{1 - \mathbf{p}_y}H(\mathbf{p}_y) + H(\tilde{\mathbf{p}}_y) + \frac{\gamma}{2}\tilde{\mathbf{p}}_y \log \tilde{\mathbf{p}}_y + \frac{\gamma}{2}(1 - \tilde{\mathbf{p}}_y)\log(1 - \tilde{\mathbf{p}}_y)$$
$$= \ell_{\boldsymbol{\theta}}(z) - \frac{1 - \tilde{\mathbf{p}}_y}{1 - \mathbf{p}_y}H(\mathbf{p}_y) + (1 - \frac{\gamma}{2})H(\tilde{\mathbf{p}}_y),$$

thus concluding the proof.

## B EFFECTS OF PERTURBATION RADIUS $\rho$ AND COEFFICIENT $\gamma$

The perturbation radius $\rho$ is an important factor in determining the generalization performance (Foret et al., 2021), but its effect on model calibration remains unknown. To answer this question, we conduct another set of experiments while varying the perturbation radius $\rho$ from 0.02 to 0.2, an interval in which the optimal value of $\rho$ is often found. Figure S5 shows that the entropy of the predictive distribution $H(\mathbf{p}_y)$ continues to increase for both models and datasets as expected. However, we also observe that for both models the test accuracy on CIFAR-10 first increases and then decreases with the perturbation radius $\rho$, though the test accuracy on CIFAR-100 keeps increasing in this interval. This implies that larger values of $\rho$ do not assure a better generalization. On the other hand, the ECE on CIFAR-10 first decreases and then increases with the perturbation radius. Moreover, the ECE of ResNet-56 is higher than that of ResNet-20 in the descending regime, which is aligned with the previous finding that increasing capacity by width or depth may hurt model calibration (Guo et al., 2017). Meanwhile, when the perturbation radius exceeds the changing point, the ECE of ResNet-20 undergoes a sudden rise and becomes higher than that of ResNet-56, a phenomenon that is more pronounced for CIFAR-10 in this interval. One explanation for this observation might be that models with low capacity are more amenable to the implicit regularization imposed by SAM. The key point is that the perturbation radius $\rho$ should be relatively small to simultaneously achieve a lower ECE and a higher test accuracy than SGD.

And below we present how the additional hyperparameter $\gamma$ of CSAM affects the final generalization and calibration. The base network is ResNet-56 trained on CIFAR-10, and the perturbation radius $\rho$ is 0.05. We sweep $\gamma$ over $\{0, 0.5, 1.0, 1.5, 2.0, 2.5, 3.0\}$ and when $\gamma = 0$, CSAM degenerates to the standard SAM. As shown in Figure S6, we can observe that when $\gamma = 0.5$, CSAM improves

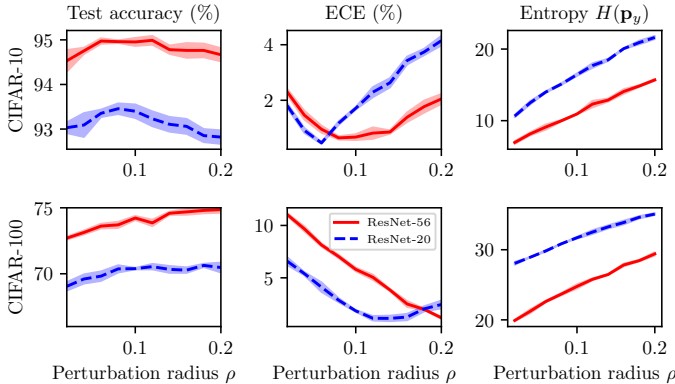

Figure S5: Variation of different metrics for models trained under monotonically increasing perturbation radius $\rho$. Note that $\mathbf{p}_y$ indicates the predicted probability associated with the true label in one-hot encoding, and $H(\mathbf{p}_y)$ is the corresponding entropy.

both the generalization and calibration. And the lowest value of ECE is attained when $\gamma = 1$, but the test accuracy slightly decreases. In contrast, increasing $\gamma$ up to 2 significantly deteriorates the performance. Therefore, a relatively smaller value of $\gamma$ is preferred.

Note that the perturbation radius $\rho$ has an important impact on $\lambda$ that controls the weight of the entropy term $-H(p_y)$. To investigate the interaction between $\rho$ and the hyper-parameter $\gamma$, we trained a number of ResNet-56 models on CIFAR-10/100 using different choices of $\rho$ and $\gamma$. Namely, $\rho$ from $\{0.05, 0.1\}$ and $\gamma$ from $\{0.0, 0.5, 1.0, 1.5, 2.0\}$. Note that when $\gamma = 0.0$, CSAM reduces to the standard SAM optimizer. From Table S4, we observe that CSAM can always generalize and calibrate better than SAM, when $\rho$ and $\gamma$ are carefully tuned. Moreover, when training with a small value of $\rho$, it is suggested to combine with a relatively large value of $\gamma$, and vice versa. This is because when $\rho$ is large, the penalty coefficient $\lambda$ in Theorem 1 is also very large. Using a large $\gamma$ in this case will over-penalize the examples, which, as a result, adversely affects the generalization and calibration.

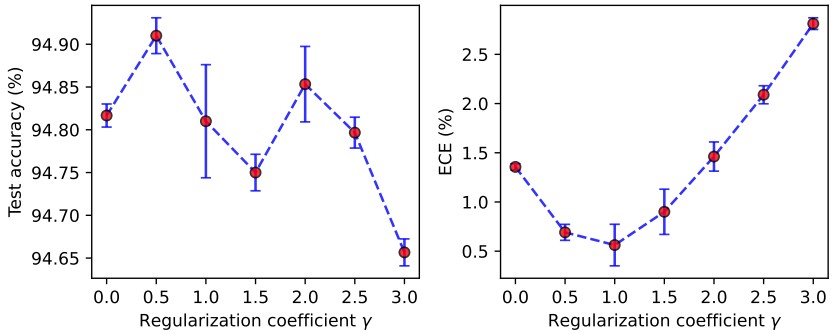

Figure S6: Effects of CSAM hyperparameter $\gamma$ on test accuracy and ECE.

## C   MORE EXPERIMENTAL RESULTS ON CSAM

In this section, we include several networks like ResNets (He et al., 2016), Wide ResNets (Zagoruyko & Komodakis, 2016), and PyramidNets (Han et al., 2017) to classify CIFAR-10/100. Furthermore, the classical ResNet-18 for ImageNet-1K is further adapted to classify Tiny-ImageNet. The initial learning rate and the weight decay coefficient are swept over $\{0.01, 0.05, 0.1\}$ and $\{1.0e$-$4, 5.0e$-$4, 1.0e$-$3\}$, respectively. By default, we use a mini-batch size of 128. The optimizer is SGD with momentum 0.9, and the learning rate is scheduled in a cosine decay (Loshchilov & Hutter, 2017). Note that all experiments are run on a GPU cluster with 2 cards, and it re-

Table S4: A number of ResNet-56 were trained on CIFAR-10/100 datasets with varying $\rho$ and $\gamma$.

| | | | $\gamma = 0.0$ | $\gamma = 0.5$ | $\gamma = 1.0$ | $\gamma = 1.5$ | $\gamma = 2.0$ |
|---|---|---|---|---|---|---|---|
| CIFAR-10 | Test Acc | $\rho = 0.05$ | $94.67 \pm 0.06$ | $94.61 \pm 0.09$ | $94.62 \pm 0.14$ | $94.67 \pm 0.14$ | $\mathbf{94.74 \pm 0.06}$ |
| | | $\rho = 0.1$ | $94.74 \pm 0.05$ | $94.71 \pm 0.07$ | $\mathbf{94.90 \pm 0.17}$ | $94.66 \pm 0.09$ | $94.79 \pm 0.05$ |
| | ECE | $\rho = 0.05$ | $1.77 \pm 0.04$ | $1.46 \pm 0.14$ | $1.09 \pm 0.07$ | $0.66 \pm 0.06$ | $\mathbf{0.65 \pm 0.14}$ |
| | | $\rho = 0.1$ | $0.82 \pm 0.12$ | $\mathbf{0.54 \pm 0.11}$ | $0.69 \pm 0.14$ | $1.34 \pm 0.33$ | $2.04 \pm 0.17$ |
| CIFAR-100 | Test Acc | $\rho = 0.05$ | $73.52 \pm 0.42$ | $73.48 \pm 0.31$ | $73.55 \pm 0.18$ | $73.57 \pm 0.46$ | $\mathbf{73.67 \pm 0.18}$ |
| | | $\rho = 0.1$ | $73.94 \pm 0.09$ | $74.16 \pm 0.17$ | $73.96 \pm 0.22$ | $\mathbf{74.12 \pm 0.09}$ | $73.86 \pm 0.39$ |
| | ECE | $\rho = 0.05$ | $8.83 \pm 0.15$ | $7.52 \pm 0.14$ | $6.24 \pm 0.28$ | $4.86 \pm 0.37$ | $\mathbf{3.67 \pm 0.12}$ |
| | | $\rho = 0.1$ | $6.17 \pm 0.18$ | $4.34 \pm 0.45$ | $3.26 \pm 0.40$ | $1.84 \pm 0.14$ | $\mathbf{1.42 \pm 0.45}$ |

quire approximately 1500 GPU hours in total. As shown from Table S6 to Table S10, CSAM consistently performs better than SAM, and it surpasses other baselines as well. The code to reproduce these results is available at `https://drive.google.com/drive/folders/1O6up8Q7sdqekErGPmetIuMfEhsPZo-Hc?usp=sharing`. While other variants of SAM potentially altered the loss landscape, their effects on calibration are supposed to be similar to SAM. This could be seen from Tables S11 and S12, which summarize the results on CIFAR-10/100. Both SAM variants, including ASAM (Kwon et al., 2021) and VASSO (Li & Giannakis, 2023), can reduce the calibration error, though ASAM generalizes much worse than SAM.

To further demonstrate the efficacy of SAM on calibration, we also compare the result against several calibration-oriented training losses such as Label Smoothing (LS) and its variants, e.g., MBLS (Liu et al., 2022a) and ACLS (Park et al., 2023). Besides them, popular approaches such as CPC (Cheng & Vasconcelos, 2022), MDCA (Hebbalaguppe et al., 2022), and CRL (Moon et al., 2020) are also included. As shown in Table S13, both Label Smoothing and its variants can reduce the ECE. But, unfortunately, they also hurt the generalization performance. In contrast, SAM and CSAM not only reduce the ECE but also significantly improve the test accuracy. Moreover, we evaluate these methods on ImageNet-1K as well. The base model is ViT-S-32 and all runs are trained only once due to limited time. Table S14 suggests that SAM/CSAM perform much better than other methods, both in generalization accuracy and calibration ECE, in ID and OOD settings.

We are also interested in whether CSAM results in a flatter minima. For this purpose, a number of ResNet-56 models are trained on CIFAR-100 with different optimizers. Table S16 indicates that CSAM can reach a flatter minima without compromising the calibration performance. Moreover, we also examine it with Focal Loss and Label Smoothing, both of which can significantly reduce the ECE value. It is interesting to find that they both converge to a flatter minima as well. This indicates a positive correlation between calibration and sharpness.

Table S5: Performance comparison between different methods on CIFAR-100. The results are averaged over 3 random seeds, with standard deviation displayed as well.

| | Test Acc ↑ | ECE ↓ | ClasswiseECE ↓ | AdaECE ↓ | TCE ↓ | AUROC ↑ |
|---|---|---|---|---|---|---|
| CE | $81.01 \pm 0.11$ | $3.95 \pm 0.28$ | $0.21 \pm 0.01$ | $3.86 \pm 0.22$ | $3.38 \pm 0.41$ | $93.93 \pm 0.05$ |
| Focal Loss (FL) | $80.55 \pm 0.17$ | $2.84 \pm 0.36$ | $0.19 \pm 0.01$ | $2.79 \pm 0.45$ | $2.75 \pm 0.36$ | $94.43 \pm 0.01$ |
| DualFocal | $80.74 \pm 0.24$ | $2.68 \pm 0.51$ | $0.18 \pm 0.01$ | $2.66 \pm 0.51$ | $2.24 \pm 0.29$ | $94.81 \pm 0.17$ |
| AdaFocal | $80.70 \pm 0.11$ | $2.58 \pm 0.31$ | $0.19 \pm 0.01$ | $2.61 \pm 0.37$ | $2.31 \pm 0.29$ | $93.75 \pm 0.05$ |
| Mixup | $82.09 \pm 0.26$ | $4.28 \pm 0.27$ | $0.18 \pm 0.02$ | $4.24 \pm 0.31$ | $4.20 \pm 0.63$ | $94.35 \pm 0.08$ |
| MIT-L | $81.29 \pm 0.18$ | $3.26 \pm 0.18$ | $0.18 \pm 0.01$ | $3.24 \pm 0.19$ | $3.09 \pm 0.49$ | $94.76 \pm 0.12$ |
| MMCE | $81.02 \pm 0.05$ | $4.02 \pm 0.29$ | $0.18 \pm 0.01$ | $3.96 \pm 0.22$ | $3.69 \pm 0.38$ | $93.83 \pm 0.07$ |
| BatchEnsemble | $79.93 \pm 0.11$ | $6.86 \pm 0.21$ | $0.21 \pm 0.01$ | $6.77 \pm 0.27$ | $2.49 \pm 0.17$ | $94.15 \pm 0.02$ |
| Rank1-BNN | $80.21 \pm 0.06$ | $3.59 \pm 0.01$ | $0.19 \pm 0.01$ | $3.57 \pm 0.08$ | $2.42 \pm 0.11$ | $94.29 \pm 0.06$ |
| VI | $76.30 \pm 0.06$ | $10.29 \pm 0.11$ | $0.27 \pm 0.03$ | $10.29 \pm 0.11$ | $2.08 \pm 0.35$ | $92.62 \pm 0.08$ |
| MIMO | $80.75 \pm 0.13$ | $2.38 \pm 0.06$ | $0.17 \pm 0.01$ | $2.31 \pm 0.04$ | $2.04 \pm 0.01$ | $95.14 \pm 0.04$ |
| ACLS | $80.49 \pm 0.19$ | $6.38 \pm 0.29$ | $0.19 \pm 0.01$ | $6.31 \pm 0.33$ | $2.90 \pm 0.47$ | $93.16 \pm 0.14$ |
| BalCAL | $81.34 \pm 0.02$ | $5.69 \pm 0.18$ | $0.18 \pm 0.01$ | $5.66 \pm 0.24$ | $2.64 \pm 0.21$ | $93.49 \pm 0.09$ |
| bSAM | $80.59 \pm 0.07$ | $8.27 \pm 0.13$ | $0.22 \pm 0.01$ | $8.27 \pm 0.14$ | $2.59 \pm 0.17$ | $94.01 \pm 0.11$ |
| SAM | $82.93 \pm 0.15$ | $2.11 \pm 0.17$ | $0.17 \pm 0.01$ | $2.17 \pm 0.21$ | $1.89 \pm 0.11$ | $94.15 \pm 0.06$ |
| CSAM | $\mathbf{83.07 \pm 0.19}$ | $\mathbf{1.93 \pm 0.15}$ | $\mathbf{0.15 \pm 0.01}$ | $\mathbf{1.99 \pm 0.05}$ | $\mathbf{1.54 \pm 0.30}$ | $\mathbf{96.07 \pm 0.03}$ |

Finally, apart from the cross-entropy loss, we are also wondering whether the calibration benefit of SAM is persistent across other training losses. As shown in Table S15, we can observe that when integrated with Focal Loss (FL) and ACLS (Park et al., 2023), SAM still calibrates better than the baseline, whereas CSAM achieves the lowest ECE/TCE.

Table S6: Results (mean±std) of test accuracy (%) over 3 random runs. Text marked as bold indicates the best result.

|  |  | CE | FL | DualFocal | AdaFocal | Mixup | MMCE | MIT-L | SAM | CSAM |
|---|---|---|---|---|---|---|---|---|---|---|
| CIFAR-10 | ResNet-56 | 94.01 ± 0.15 | 93.99 ± 0.04 | 93.84 ± 0.22 | 93.87 ± 0.08 | 94.42 ± 0.15 | 94.19 ± 0.22 | 94.68 ± 0.08 | 94.92 ± 0.24 | **95.00 ± 0.25** |
|  | WRN-28-10 | 95.83 ± 0.21 | 95.91 ± 0.02 | 95.73 ± 0.10 | 95.78 ± 0.06 | 96.64 ± 0.10 | 95.94 ± 0.02 | 96.56 ± 0.16 | **96.91± 0.14** | 96.87 ± 0.05 |
|  | PyramidNet-110 | 96.07 ± 0.23 | 96.03 ± 0.06 | 96.14 ± 0.04 | 96.00 ± 0.11 | 96.77 ± 0.08 | 96.13 ± 0.08 | 96.78 ± 0.17 | 97.14 ± 0.06 | **97.26 ± 0.03** |
| CIFAR-100 | ResNet-56 | 72.06 ± 0.13 | 71.96 ± 0.28 | 71.43 ± 0.04 | 72.00 ± 0.08 | 74.15 ± 0.29 | 72.17 ± 0.12 | 74.28 ± 0.42 | 74.71 ± 0.30 | **74.95 ± 0.32** |
|  | WRN-28-10 | 81.04 ± 0.11 | 80.55 ± 0.17 | 80.74 ± 0.24 | 80.70 ± 0.11 | 82.09 ± 0.26 | 81.02 ± 0.05 | 81.29 ± 0.18 | 82.93 ± 0.15 | **83.05 ± 0.19** |
|  | PyramidNet-110 | 81.21 ± 0.52 | 81.53 ± 0.12 | 81.76 ± 0.07 | 81.81 ± 0.38 | 82.94 ± 0.29 | 81.36 ± 0.31 | 82.41 ± 0.02 | 84.08 ± 0.29 | **84.16 ± 0.15** |
| Tiny-ImageNet | ResNet-18 | 51.96 ± 0.35 | 52.61 ± 0.59 | 53.02 ± 0.86 | 50.36 ± 0.69 | 51.45 ± 0.70 | 51.31 ± 0.79 | 51.97 ± 0.24 | 56.81 ± 0.31 | **57.13 ± 0.96** |

Table S7: Results (mean±std) of ECE (%) with $M = 15$ over 3 random runs. Text marked as bold indicates the best result.

|  |  | CE | FL | DualFocal | AdaFocal | Mixup | MMCE | MIT-L | SAM | CSAM |
|---|---|---|---|---|---|---|---|---|---|---|
| CIFAR-10 | ResNet-56 | 3.89 ± 0.16 | 1.81 ± 0.12 | 2.50 ± 0.03 | 0.89 ± 0.12 | 3.87 ± 0.09 | 3.61 ± 0.17 | 1.83 ± 0.18 | 0.64 ± 0.09 | **0.58 ± 0.07** |
|  | WRN-28-10 | 2.36 ± 0.11 | 1.16 ± 0.13 | 4.74 ± 0.09 | 0.91 ± 0.14 | 4.66 ± 1.11 | 2.47 ± 0.04 | 1.05 ± 0.02 | 0.86 ± 0.13 | **0.50 ± 0.03** |
|  | PyramidNet-110 | 2.54 ± 0.19 | 1.17 ± 0.15 | 4.64 ± 0.05 | 0.96 ± 0.12 | 2.23 ± 0.84 | 2.49 ± 0.12 | 1.22 ± 0.14 | 0.74 ± 0.08 | **0.32 ± 0.06** |
| CIFAR-100 | ResNet-56 | 13.29 ± 0.15 | 8.25 ± 0.23 | 4.93 ± 0.06 | 1.71 ± 0.09 | 2.43 ± 0.32 | 13.49 ± 0.19 | 5.11 ± 1.38 | 1.66 ± 0.16 | **0.84 ± 0.15** |
|  | WRN-28-10 | 3.95 ± 0.28 | 2.84 ± 0.36 | 12.66 ± 0.51 | 2.58 ± 0.31 | 4.28 ± 0.27 | 4.02 ± 0.29 | 3.26 ± 0.18 | 2.11 ± 0.17 | **1.50 ± 0.07** |
|  | PyramidNet-110 | 9.52 ± 0.64 | 4.26 ± 0.39 | 10.58 ± 0.55 | 1.95 ± 0.11 | 3.25 ± 1.19 | 9.24 ± 0.38 | 3.03 ± 0.38 | 1.91 ± 0.14 | **1.69 ± 0.04** |
| Tiny-ImageNet | ResNet-18 | 7.65 ± 2.21 | 4.35 ± 0.64 | 16.30 ± 0.53 | 11.71 ± 0.66 | 10.81 ± 0.66 | 9.34 ± 2.10 | 4.09 ± 0.17 | 3.46 ± 0.15 | **2.75 ± 0.47** |

Table S8: Results (mean±std) of Classwise ECE (%) with $M = 15$ over 3 random runs. Text marked as bold indicates the best result.

|  |  | CE | FL | DualFocal | AdaFocal | Mixup | MMCE | MIT-L | SAM | CSAM |
|---|---|---|---|---|---|---|---|---|---|---|
| CIFAR-10 | ResNet-56 | 0.80 ± 0.01 | 0.47 ± 0.02 | 0.67 ± 0.01 | 0.37 ± 0.03 | 0.87 ± 0.06 | 0.78 ± 0.03 | 0.46 ± 0.02 | 0.32 ± 0.01 | **0.29 ± 0.02** |
|  | WRN-28-10 | 0.52 ± 0.01 | 0.38 ± 0.01 | 1.11 ± 0.02 | 0.35 ± 0.00 | 1.05 ± 0.23 | 0.54 ± 0.02 | 0.31 ± 0.01 | 0.26 ± 0.02 | **0.23 ± 0.01** |
|  | PyramidNet-110 | 0.56 ± 0.03 | 0.36 ± 0.02 | 1.03 ± 0.03 | 0.34 ± 0.01 | 0.45 ± 0.02 | 0.55 ± 0.02 | 0.31 ± 0.01 | 0.25 ± 0.01 | **0.20 ± 0.01** |
| CIFAR-100 | ResNet-56 | 0.32 ± 0.01 | 0.25 ± 0.00 | 0.21 ± 0.01 | 0.19 ± 0.00 | 0.19 ± 0.00 | 0.33 ± 0.00 | 0.20 ± 0.01 | **0.16 ± 0.00** | **0.16 ± 0.00** |
|  | WRN-28-10 | 0.18 ± 0.01 | 0.19 ± 0.00 | 0.34 ± 0.01 | 0.19 ± 0.00 | 0.19 ± 0.00 | 0.18 ± 0.01 | 0.18 ± 0.01 | 0.17 ± 0.00 | **0.15 ± 0.01** |
|  | PyramidNet-110 | 0.23 ± 0.01 | 0.17 ± 0.00 | 0.30 ± 0.01 | 0.17 ± 0.00 | 0.18 ± 0.03 | 0.23 ± 0.01 | 0.16 ± 0.00 | 0.15 ± 0.00 | **0.14 ± 0.01** |
| Tiny-ImageNet | ResNet-18 | 0.21 ± 0.01 | **0.19 ± 0.00** | 0.24 ± 0.01 | 0.23 ± 0.01 | 0.21 ± 0.01 | 0.21 ± 0.01 | **0.19 ± 0.00** | **0.19 ± 0.00** | **0.19 ± 0.00** |

Table S9: Results (mean±std) of Adaptive ECE (%) with $M = 15$ over 3 random runs. Text marked as bold indicates the best result.

| | | CE | FL | DualFocal | AdaFocal | Mixup | MMCE | MIT-L | SAM | CSAM |
|---|---|---|---|---|---|---|---|---|---|---|
| CIFAR-10 | ResNet-56 | 3.71 ± 0.06 | 2.13 ± 0.19 | 2.19 ± 0.18 | 1.03 ± 0.14 | 3.97 ± 0.08 | 3.55 ± 0.15 | 1.83 ± 0.14 | 0.90 ± 0.14 | **0.51 ± 0.03** |
| | WRN-28-10 | 2.36 ± 0.11 | 1.42 ± 0.09 | 4.64 ± 0.07 | 0.65 ± 0.04 | 4.63 ± 1.04 | 2.42 ± 0.04 | 1.05 ± 0.05 | 0.84 ± 0.14 | **0.48 ± 0.04** |
| | PyramidNet-110 | 2.53 ± 0.19 | 1.78 ± 0.07 | 4.54 ± 0.02 | 0.88 ± 0.09 | 2.69 ± 0.22 | 2.49 ± 0.13 | 1.19 ± 0.17 | 0.70 ± 0.05 | **0.19 ± 0.02** |
| CIFAR-100 | ResNet-56 | 13.36 ± 0.12 | 8.23 ± 0.26 | 4.91 ± 0.06 | 1.82 ± 0.21 | 2.48 ± 0.23 | 13.48 ± 0.21 | 5.09 ± 1.36 | 1.02 ± 0.02 | **0.96 ± 0.14** |
| | WRN-28-10 | 3.86 ± 0.22 | 2.79 ± 0.45 | 12.66 ± 0.51 | 2.61 ± 0.37 | 4.24 ± 0.31 | 3.96 ± 0.22 | 3.24 ± 0.19 | 4.67 ± 0.21 | **1.50 ± 0.01** |
| | PyramidNet-110 | 9.29 ± 0.54 | 4.06 ± 0.49 | 10.58 ± 0.55 | 1.76 ± 0.22 | 3.25 ± 1.01 | 9.18 ± 0.41 | 2.99 ± 0.02 | 1.65 ± 0.14 | **1.45 ± 0.04** |
| Tiny-ImageNet | ResNet-18 | 7.55 ± 2.27 | 4.25 ± 0.56 | 16.31 ± 0.54 | 11.71 ± 0.66 | 10.79 ± 0.64 | 9.19 ± 2.12 | 3.33 ± 0.14 | 4.07 ± 0.19 | **2.65 ± 0.30** |

Table S10: Results (mean±std) of AUROC (%) over 3 random runs. Text marked as bold indicates the best result.

| | | CE | FL | DualFocal | AdaFocal | Mixup | MMCE | MIT-L | SAM | CSAM |
|---|---|---|---|---|---|---|---|---|---|---|
| CIFAR-10 | ResNet-56 | 97.98 ± 0.03 | 98.47 ± 0.05 | 98.73 ± 0.09 | 98.78 ± 0.03 | 98.59 ± 0.03 | 98.04 ± 0.02 | 98.72 ± 0.03 | 99.07 ± 0.07 | **99.19 ± 0.02** |
| | WRN-28-10 | 98.68 ± 0.04 | 99.04 ± 0.01 | 99.26 ± 0.02 | 99.10 ± 0.04 | 99.12 ± 0.02 | 98.65 ± 0.05 | 99.12 ± 0.03 | 99.30 ± 0.02 | **99.40 ± 0.01** |
| | PyramidNet-110 | 98.64 ± 0.04 | 98.96 ± 0.04 | 99.40 ± 0.04 | 99.16 ± 0.04 | 99.00 ± 0.04 | 98.66 ± 0.04 | 99.20 ± 0.02 | 99.41 ± 0.03 | **99.52 ± 0.02** |
| CIFAR-100 | ResNet-56 | 91.06 ± 0.01 | 92.32 ± 0.09 | 92.69 ± 0.09 | 93.44 ± 0.07 | 92.87 ± 0.09 | 90.99 ± 0.04 | 93.32 ± 0.27 | 94.35 ± 0.05 | **94.57 ± 0.07** |
| | WRN-28-10 | 93.93 ± 0.05 | 94.43 ± 0.01 | 94.81 ± 0.17 | 93.75 ± 0.05 | 94.35 ± 0.08 | 93.83 ± 0.07 | 94.76 ± 0.12 | 94.15 ± 0.06 | **96.06 ± 0.03** |
| | PyramidNet-110 | 93.46 ± 0.22 | 94.29 ± 0.01 | 95.16 ± 0.03 | 95.03 ± 0.09 | 94.62 ± 0.04 | 93.47 ± 0.13 | 95.21 ± 0.08 | 96.05 ± 0.07 | **96.14 ± 0.05** |
| Tiny-ImageNet | ResNet-18 | 82.62 ± 0.37 | 83.31 ± 0.63 | 81.08 ± 0.42 | 81.32 ± 0.65 | 79.32 ± 0.53 | 82.57 ± 0.06 | 82.38 ± 0.13 | 85.63 ± 0.31 | **85.69 ± 0.15** |

Table S11: Calibration performance of different SAM variants on CIFAR-10.

| | Test Acc | ECE | ClasswiseECE | AdaECE | TCE | AUROC |
|---|---|---|---|---|---|---|
| SGD | 94.26 ± 0.06 | 3.52 ± 0.12 | 0.78 ± 0.01 | 3.52 ± 0.11 | 1.03 ± 0.01 | 98.09 ± 0.08 |
| SAM | 94.50 ± 0.20 | 1.83 ± 0.08 | 0.44 ± 0.01 | 1.78 ± 0.03 | 0.70 ± 0.12 | 98.78 ± 0.08 |
| ASAM | 94.82 ± 0.11 | 2.03 ± 0.11 | 0.48 ± 0.01 | 2.01 ± 0.14 | 0.61 ± 0.05 | 98.71 ± 0.02 |
| VASSO | 94.70 ± 0.10 | 1.75 ± 0.04 | 0.42 ± 0.01 | 1.69 ± 0.05 | 0.71 ± 0.09 | 98.87 ± 0.06 |
| CSAM | **94.58 ± 0.14** | **1.47 ± 0.17** | **0.41 ± 0.02** | **1.41 ± 0.19** | **0.72 ± 0.20** | **98.87 ± 0.06** |

Table S12: Calibration performance of different SAM variants on CIFAR-100.

| | Test Acc | ECE | ClasswiseECE | AdaECE | TCE | AUROC |
|---|---|---|---|---|---|---|
| SGD | 72.00 ± 0.15 | 13.14 ± 0.25 | 0.33 ± 0.01 | 13.13 ± 0.25 | 1.64 ± 0.09 | 91.12 ± 0.11 |
| SAM | 74.87 ± 0.21 | 1.59 ± 0.10 | 0.17 ± 0.01 | 1.39 ± 0.16 | 1.38 ± 0.11 | 94.40 ± 0.01 |
| ASAM | 74.11 ± 0.11 | 6.48 ± 0.26 | 0.22 ± 0.01 | 6.43 ± 0.25 | 1.31 ± 0.31 | 93.13 ± 0.09 |
| VASSO | 74.94 ± 0.53 | 1.55 ± 0.25 | 0.17 ± 0.01 | 1.55 ± 0.24 | 1.43 ± 0.23 | 94.31 ± 0.07 |
| CSAM | **74.85 ± 0.20** | **1.31 ± 0.40** | **0.17 ± 0.01** | **1.22 ± 0.23** | **1.36 ± 0.44** | **94.55 ± 0.06** |

Table S13: Comparison against other calibration-oriented training losses.

| | SGD | LS | ACLS | CRL | CPC | MBLS | MDCA | SAM | CSAM |
|---|---|---|---|---|---|---|---|---|---|
| Test Acc | 72.30 ± 0.12 | 72.54 ± 0.18 | 72.49 ± 0.05 | 72.36 ± 0.18 | 72.45 ± 0.47 | 72.42 ± 0.24 | 72.38 ± 0.48 | 74.63 ± 0.24 | **74.98 ± 0.16** |
| ECE | 13.03 ± 0.26 | 4.38 ± 0.14 | 2.32 ± 0.40 | 3.43 ± 0.25 | 2.62 ± 0.54 | 2.33 ± 0.42 | 3.16 ± 0.17 | 1.87 ± 0.25 | **1.25 ± 0.15** |

Table S14: Results on the ImageNet-1K dataset. Slightly different from the custom setting, we reserve 20% of the ImageNet-1K validation set as a new validation set for early stopping and temperature scaling, and the remaining images therefore constitute a test set. Both metrics are evaluated on the test set (1/2/3 indicate different types of corruption).

| | **ID Metrics** | | **OOD Metrics** | | | | | |
|---|---|---|---|---|---|---|---|---|
| | Test Acc ↑ | ECE ↓ | Test Acc (1/2/3) ↑ | | | ECE (1/2/3) ↓ | | |
| SGD | 65.91 | 11.56 | 33.54 | 30.21 | 34.18 | 23.70 | 24.65 | 18.00 |
| MBLS | 67.19 | 2.21 | 34.54 | 31.58 | 35.58 | 11.83 | 12.61 | 8.58 |
| CPC | 65.68 | 9.34 | 32.34 | 31.01 | 35.71 | 20.46 | 23.19 | 15.04 |
| ACLS | 66.68 | 3.97 | 34.67 | 31.88 | 36.00 | 15.72 | 16.17 | 10.25 |
| MDCA | 65.23 | 8.38 | 33.11 | 29.91 | 33.71 | 23.31 | 24.28 | 17.15 |
| SAM | 69.48 | 2.47 | 38.25 | 35.27 | 39.54 | 7.33 | 7.77 | 3.15 |
| CSAM | **69.78** | **1.79** | **38.58** | **35.46** | **39.85** | **7.07** | **7.70** | **2.76** |

Table S15: Performance of CSAM/SAM when combined with other training losses (e.g. FL and ACLS).

| | ResNet-56 | | | WRN-28-10 | | |
|---|---|---|---|---|---|---|
| | Test Acc ↑ | ECE ↓ | TCE ↓ | Test Acc ↑ | ECE ↓ | TCE ↓ |
| FL | $71.96 \pm 0.28$ | $8.25 \pm 0.23$ | $3.27 \pm 0.16$ | $80.55 \pm 0.17$ | $2.84 \pm 0.36$ | $2.75 \pm 0.36$ |
| FL+SAM | $73.11 \pm 0.01$ | $6.90 \pm 0.38$ | $1.90 \pm 0.13$ | $81.33 \pm 0.23$ | $2.41 \pm 0.06$ | $1.76 \pm 0.12$ |
| FL+CSAM | $\mathbf{73.51 \pm 0.10}$ | $\mathbf{3.94 \pm 0.17}$ | $\mathbf{1.58 \pm 0.13}$ | $\mathbf{82.08 \pm 0.08}$ | $\mathbf{1.88 \pm 0.11}$ | $\mathbf{1.42 \pm 0.16}$ |
| ACLS | $72.55 \pm 0.08$ | $5.88 \pm 0.18$ | $3.29 \pm 0.25$ | $80.49 \pm 0.19$ | $6.38 \pm 0.29$ | $2.90 \pm 0.47$ |
| ACLS+SAM | $74.53 \pm 0.09$ | $1.37 \pm 0.05$ | $1.27 \pm 0.18$ | $83.04 \pm 0.01$ | $1.87 \pm 0.10$ | $1.80 \pm 0.12$ |
| ACLS+CSAM | $\mathbf{74.86 \pm 0.07}$ | $\mathbf{1.01 \pm 0.12}$ | $\mathbf{0.85 \pm 0.05}$ | $\mathbf{83.13 \pm 0.08}$ | $\mathbf{1.37 \pm 0.21}$ | $\mathbf{1.36 \pm 0.06}$ |

Table S16: Comparison of sharpness (measured by the largest eigenvalue $\kappa_{max}$ of the Hessian) of different optimizers.

| | SGD | Focal Loss | Label Smoothing | SAM | CSAM |
|---|---|---|---|---|---|
| ECE | $13.03 \pm 0.26$ | $1.85 \pm 0.09$ | $2.31 \pm 0.15$ | $1.27 \pm 0.25$ | $\mathbf{1.08 \pm 0.12}$ |
| $\kappa_{max}$ | $631.89 \pm 86.81$ | $496.99 \pm 22.02$ | $583.74 \pm 52.08$ | $177.99 \pm 13.25$ | $\mathbf{148.16 \pm 10.09}$ |

