# OpenReview forum: "Towards Understanding The Calibration Benefits of Sharpness-Aware Minimization"
_ICLR.cc/2026/Conference — ICLR 2026 Poster_

### Official Review · Reviewer_7xpc · 2025-10-26

**Soundness:** 3
**Presentation:** 3
**Contribution:** 2
**Rating:** 2
**Confidence:** 5

**Summary:**

This paper investigates why Sharpness-Aware Minimization (SAM) tends to produce better-calibrated neural networks than standard optimizers such as SGD and AdamW.
The authors provide theoretical analysis showing that SAM implicitly regularizes the negative entropy of the predictive distribution, similar to focal loss, and thus discourages over-confident predictions.
Building on this, they propose a simple variant Calibrated SAM (CSAM), which re-weights over-confident examples by modifying the SAM outer loss.
Experiments on CIFAR-10/100 and ImageNet-1K demonstrate that (1) SAM improves Expected Calibration Error (ECE) compared with SGD and common post-hoc methods, and (2) CSAM further improves calibration without harming accuracy.

**Strengths:**

1. Provides a clean theoretical link between sharpness minimization and confidence calibration.
2. Solid empirical confirmation that SAM reduces calibration error across multiple architectures.
3. The CSAM variant is simple to implement and empirically effective.
4. Clear exposition and well-structured proofs.

**Weaknesses:**

1. The core theoretical insight—that SAM implicitly maximizes predictive entropy—is conceptually close to prior entropy-regularization or focal-loss analyses; novelty is limited.
2. The proposed CSAM is a minimal heuristic extension rather than a principled optimization method.
3. Missing systematic comparisons against other calibration-oriented training losses such as label smoothing. Although table3 is given but the setting is very limited. Author should also test if the CSAM be compatible with other training loss.
4. Analysis of how calibration evolves during training (claimed in Appendix D) is not deeply discussed in the main text.

**Questions:**

Would the effect persist if temperature scaling or focal loss were combined with SAM?

---

> ### Author Response · Authors · 2025-11-19
>
> Many thanks for the valuable comments, and they are addressed one by one as follows.
>
>  (**Q1**): The core theoretical insight—that SAM implicitly maximizes predictive entropy—is conceptually close to prior entropy-regularization or focal-loss analyses; novelty is limited.
>
> (**R1**): Thanks for the comment. In many cases, either entropy-regularization or focal loss can indeed improve calibration, although often at the cost of generalization. In contrast, SAM, as a novel optimizer, can significantly improve both generalization and calibration. This is critically important to the optimization community. However, most studies only focus on explaining why SAM significantly generalizes better than SGD, and little work has been devoted to studying its calibration benefits. It should be noted that generalization and calibration are not linearly correlated: a well-calibrated network does not indicate a well-generalizable network, and vice versa. Therefore, by first building the connection between SAM and maximizing predictive entropy, we
> - (theoretically) address the issue of why SAM can improve calibration
> - (practically) provide new directions to improve SAM for better generalization and calibration.
>
> And as a result, the interpretation from the perspective of maximizing predictive entropy would be highly valuable to the community.
>
> (**Q2**): The proposed CSAM is a minimal heuristic extension rather than a principled optimization method.
>
> (**R2**): Thanks for the comment. We understand that you might think that the modification of the loss function is minor. However, it should be clarified that CSAM is widely applicable and well theoretically supported (see Theorem 3). By introducing an additional parameter $\gamma$, CSAM is more flexible than SAM to balance generalization and calibration. For example, the table below (data from Table S4) shows that CSAM can significantly improve calibration by choosing a proper $\gamma$. Moreover, in the sequel, we also demonstrate that CSAM can be integrated with other training losses, which further shows that CSAM is a principled optimization method for better calibration and generalization.
>
> |           	|          	|             	| $\gamma=0.0$     	| $\gamma=0.5$     	| $\gamma=1.0$     	| $\gamma=1.5$     	| $\gamma=2.0$     	|
> |---------|-----|-----------|----------------|----------------|--------------|----------------|-----------------|
> |      CIFAR-10       	| ECE      	| $\rho=0.05$ 	| 1.77 $\pm$ 0.04  	| 1.46 $\pm$ 0.14  	| 1.09 $\pm$ 0.07  	| 0.66 $\pm$ 0.06  	| **0.65 $\pm$ 0.14(-63.28%)**  	|
> |           	|          	| $\rho=0.1$  	| 0.82 $\pm$ 0.12  	| **0.54 $\pm$ 0.11(-34.15%)**  	| 0.69 $\pm$ 0.14  	| 1.34 $\pm$ 0.33  	| 2.04 $\pm$ 0.17  	|
> |     CIFAR-100       	| ECE      	| $\rho=0.05$ 	| 8.83 $\pm$ 0.15  	| 7.52$\pm$ 0.14   	| 6.24 $\pm$ 0.28  	| 4.86 $\pm$ 0.37  	| **3.67 $\pm$ 0.12(-58.43%)**  	|
> |           	|          	| $\rho=0.1$  	| 6.17 $\pm$ 0.18  	| 4.34 $\pm$ 0.45  	| 3.26 $\pm$ 0.40  	| 1.84 $\pm$ 0.14  	| **1.42 $\pm$ 0.45(-76.98%)**  	|
>
>
> (**Q3**):  Missing systematic comparisons against other calibration-oriented training losses such as label smoothing. Although table3 is given but the setting is very limited. Author should also test if the CSAM be compatible with other training loss.
>
> (**R3**): Thanks for pointing out this issue. First, we added several calibration-oriented training losses as baselines, such as Label Smoothing (LS) and its variants, e.g., MBLS (Liu et al., 2022) and ACLS (Park et al., 2023), and several popular approaches, such as CPC (Cheng et al., 2022), MDCA (hebbalaguppe et al., 2022), and CRL (Moon et al., 2020) are also included. As shown in the following Table, both Label Smoothing and its variants can reduce the ECE. But, unfortunately, they also hurt the generalization performance. In contrast, SAM and CSAM not only reduce the ECE but also significantly improve the test accuracy.
>
> |   |SGD       |        LS        |ACLS             |CRL              |CPC              | MBLS             |MDCA           |SAM          |CSAM     |
> |--|-------|-----------|-----------------|-----------------|-------------|-----------------------|---------------|--------------|------------|
> | Test Acc | 72.30 $\pm$ 0.12 | 72.54 $\pm$ 0.18 | 72.49 $\pm$ 0.05 | 72.36 $\pm$ 0.18 | 72.45 $\pm$ 0.47 | 72.42 $\pm$ 0.24 | 72.38 $\pm$ 0.48 | 74.63 $\pm$ 0.24 | **74.98 $\pm$ 0.16** 	|
> | ECE | 13.03 $\pm$ 0.26 | 4.38 $\pm$ 0.14  | 2.32 $\pm$ 0.40  | 3.43 $\pm$ 0.25  | 2.62 $\pm$ 0.54  | 2.33 $\pm$ 0.42  | 3.16 $\pm$ 0.17  | 1.87 $\pm$ 0.25  | **1.25 $\pm$ 0.15**  |

---

> ### Author Response · Authors · 2025-11-19
> **Continued**
>
> Moreover, we evaluate these methods on ImageNet-1K as well. The base model is ViT-S-32. The Table below suggests that SAM/CSAM perform much better than other methods, both in generalization accuracy and calibration ECE, in ID and OOD settings.
> |      	|      ID Metrics     	|                  	|         OOD Metrics         	|       	|       	|                          	|       	|       	|
> |-----|------------------|---------------|--------------------------|----|----|-----------------------|-----|----|
> |      	| Test Acc $\uparrow$ 	| ECE $\downarrow$ 	| Test Acc (1/2/3) $\uparrow$ 	|       	|       	| ECE (1/2/3) $\downarrow$ 	|       	|       	|
> | SGD  	|        65.91        	|       11.56      	|            33.54            	| 30.21 	| 34.18 	|           23.70          	| 24.65 	| 18.00 	|
> | MBLS 	|        67.19        	|       2.21       	|            34.54            	| 31.58 	| 35.58 	|           11.83          	| 12.61 	|  8.58 	|
> | CPC  	|        65.68        	|       9.34       	|            32.34            	| 31.01 	| 35.71 	|           20.46          	| 23.19 	| 15.04 	|
> | ACLS 	|        66.68        	|       3.97       	|            34.67            	| 31.88 	| 36.00 	|           15.72          	| 16.17 	| 10.25 	|
> | MDCA 	|        65.23        	|       8.38       	|            33.11            	| 29.91 	| 33.71 	|           23.31          	| 24.28 	| 17.15 	|
> | SAM  	|        69.48        	|       2.47       	|            38.25            	| 35.27 	| 39.54 	|           7.33           	|  7.77 	|  3.15 	|
> | CSAM 	|        **69.78**        	|       **1.79**       	|            **38.58**            	| **35.46** 	| **39.85** 	|           **7.07**           	|  **7.70** 	|  **2.76** 	|
>
> Furthermore, we also test whether CSAM is compatible with other training losses. As examples, we use Focal Loss (FL) and ACLS (Park et al., 2023) as two baselines. As the table below shows, we can observe that SAM calibrates better than the baselines, and CSAM performs even better than SAM. This result further demonstrates the efficacy of CSAM/SAM in improving calibration. **All these revisions can be found in Tables S13, S14, and S15**.
>
> |           	|      ResNet-56      	|                  	|                  	|      WRN-28-10      	|                  	|                  	|
> |----------|----------------|-------------|-----------|---------------|-------------|---------------|
> |           	| Test Acc $\uparrow$ 	| ECE $\downarrow$ 	| TCE $\downarrow$ 	| Test Acc $\uparrow$ 	| ECE $\downarrow$ 	| TCE $\downarrow$ 	|
> | FL        	|   $71.96 \pm 0.28$  	|  $8.25 \pm 0.23$ 	|  $3.27 \pm 0.16$ 	|   $80.55 \pm 0.17$  	|  $2.84 \pm 0.36$ 	|  $2.75 \pm 0.36$ 	|
> | FL+SAM    	|   $73.11 \pm 0.01$  	|  $6.90 \pm 0.38$ 	|  $1.90 \pm 0.13$ 	|   $81.33 \pm 0.23$  	|  $2.41 \pm 0.06$ 	|  $1.76 \pm 0.12$ 	|
> | FL+CSAM   	|   **73.51 $\pm$ 0.10**  	|  **3.94 $\pm$ 0.17** 	|  **1.58 $\pm$ 0.13** 	|   **82.08 $\pm$ 0.08**  	|  **1.88 $\pm$ 0.11** 	|  **1.42 $\pm$ 0.16** 	|
> | ACLS      	|   $72.55 \pm 0.08$  	|  $5.88 \pm 0.18$ 	|  $3.29 \pm 0.25$ 	|   $80.49 \pm 0.19$  	|  $6.38 \pm 0.29$ 	|  $2.90 \pm 0.47$ 	|
> | ACLS+SAM  	|   $74.53 \pm 0.09$  	|  $1.37 \pm 0.05$ 	|  $1.27 \pm 0.18$ 	|   $83.04 \pm 0.01$  	|  $1.87 \pm 0.10$ 	|  $1.80 \pm 0.12$ 	|
> | ACLS+CSAM 	|   **74.86 $\pm$ 0.07**  	|  **1.01 $\pm$ 0.12** 	|  **0.85 $\pm$ 0.05** 	|   **83.13 $\pm$ 0.08**  	|  **1.37 $\pm$ 0.21** 	|  **1.36 $\pm$ 0.06** 	|
>
> - Cheng et al., Calibrating deep neural networks by pairwise constraints. CVPR 2022.
> - Hebbalaguppe et al., A stitch in time saves nine: A train-time regularizing loss for improved neural network calibration. CVPR 2022.
> - Moon et al., Confidence-aware learning for deep neural networks. ICML 2020.
> - Liu et al., Margin-based label smoothing for network calibration. CVPR 2022.
> - Park et al., ACLS: adaptive and conditional label smoothing for network calibration. ICCV 2023.
>
> (**Q4**):  Analysis of how calibration evolves during training (claimed in Appendix D) is not deeply discussed in the main text.
>
> (**R4**): Thanks for the kind reminder. In the revision, we have added the figure (see Figure 3c-d) to monitor how the ECE and TCE (namely, ECE calibrated by temperature scaling). The results suggest that SAM always achieves lower ECE/TCE than SGD.
>
> (**Q5**):  Would the effect persist if temperature scaling or focal loss were combined with SAM?
>
> (**R5**): Thanks for the comment. Actually, as you may see in Figure 3c-d, the effect is persistent even when temperature scaling is further applied. Moreover, as shown by the last table of **(Q3)**, we can also observe that the effect persists when the focal loss is combined with SAM. All the revisions can be found in Figure 3 and Table S15.
>
>
> At last, many thanks for your time and effort. Please let us know if you have further questions. And if you are satisfied with our response, please kindly raise the score.

---

> > ### Comment · Reviewer_7xpc · 2025-11-25
> >
> > Thank you for providing the additional experiments and extended comparisons, the empirical section is now much stronger.
> >
> > However, my concern about the theoretical novelty remains. The core idea that SAM improves calibration by implicitly encouraging higher predictive entropy is still conceptually close to prior analyses of entropy-regularization and focal-loss–based calibration (e.g., Calibrating Deep Neural Networks using Focal Loss), which analyze very similar mechanisms.
> >
> > Therefore, I still believe the theoretical contribution is limited.
> >
> > Regarding CSAM, although the new results show it is compatible with various losses, it still feels like a heuristic extension of SAM rather than a principled optimization method.

---

> > > ### Author Response · Authors · 2025-11-25
> > >
> > > Thanks for the comment. Unfortunately, we can not agree with your opinion and the reasons are as follows:
> > > - First, **the connection between SAM optimizer and maximizing predictive entropy is not trivial**. Methods such as entropy regularization and focal loss only modifies the objective function, while SAM optimizer proposes a fresh paradigm of learning. **As a result, entropy regularization often reduces the calibration error at the cost of generalization**. In constrat, SAM improves both calibration and generalization. By building the connection between SAM and maximizing predictive entropy, we provide new insights in studying the underlying mechanism of SAM. This finding has several important implications. For example, it prompts us to investigate whether the class of sharpness-based optimizers such as    Entropy-SGD or LPF-SGD have similar calibration benefits. Also, it lends hand to the study of the relation between sharpness, calibration, and generalization.
> > >
> > > - Second, **the many experiments demonstrate that CSAM is scalable, useful and compatible in different tasks**. So, while our primary purpose is not to propose a novel optimizer, CSAM remains to be an effective and theoretically-grounded improvement of SAM.

---

> > > > ### Comment · Reviewer_7xpc · 2025-11-28
> > > >
> > > > Thank you for the reply. The explanation on the non-trivialness of the connection between focal loss and SAM make sense. I will increase my socre to 6.

---

> > > > > ### Author Response · Authors · 2025-11-28
> > > > >
> > > > > Dear Reviewer 7xpc,
> > > > >
> > > > > We are glad to hear that your concerns are addressed. Please let us know if you have further questions.
> > > > >
> > > > > Best wishes!

---

### Official Review · Reviewer_AfCQ · 2025-10-31

**Soundness:** 3
**Presentation:** 3
**Contribution:** 2
**Rating:** 4
**Confidence:** 4

**Summary:**

This study primarily explores the relationship between the probability-based loss output at the point of maximum loss within the perturbation radius 𝜌 and the loss-to-probability mapping under the current model parameters. This analysis establishes a theoretical connection between model calibration and Sharpness-Aware Minimization (SAM), elucidating their underlying mechanisms. Building upon these findings, the authors propose an improved method aimed at enhancing model calibration.

**Strengths:**

1. Both SAM and model calibration represent important research directions, and exploring their mutual influence holds significant scientific value.
2. The study demonstrates strong completeness by beginning with theoretical foundations to explain their respective roles and subsequently extending the analysis to methodological applications.

**Weaknesses:**

1. Although Theorems 1, 2, and 3 are presented, they largely reflect variations of similar concepts. Consequently, the theoretical framework appears somewhat repetitive and lacks sufficient depth.
2. The methodological contribution is not particularly innovative, as it mainly combines the existing SAM framework with focal loss. Thus, the degree of novelty—especially from a methodological perspective—may be limited in its ability to inspire readers.

**Questions:**

1. Table S4 shows improved calibration results when 𝛾=0. Does this suggest that the proposed enhancement does not significantly improve calibration performance? At  𝛾=0, the method reduces to the standard SAM framework.
2. The conclusion states: “We proved that SAM achieves this goal by imposing an implicit regularization on the negative entropy of the predictive distribution during training, which is similar to focal loss.” Could the authors clarify which mathematical expression supports this statement? No corresponding equation appears in the theoretical derivations or references. A more detailed explanation would be appreciated.
3. Since the proposed method is derived from the theoretical analysis, can the assumptions established in the theory be satisfied in practical applications? In other words, do the conclusions derived under these assumptions hold in real-world optimization scenarios?

---

> ### Author Response · Authors · 2025-11-19
>
> Many thanks for the valuable comments, and they are addressed one by one as follows.
>
>  (**Q1**): Although Theorems 1, 2, and 3 are presented, they largely reflect variations of similar concepts. Consequently, the theoretical framework appears somewhat repetitive and lacks sufficient depth.
>
> (**R1**): Thanks for the comment. Actually, Theorem 1 argues that SAM implicitly maximizes the predictive entropy $H(\mathbf{p} _ y)$ during training, whereas Theorem 3 argues that the proposed optimizer CSAM can further magnify this effect. In the meantime, Theorem 2 is intended to demonstrate that the argument also holds for the realistic mini-batch case. While both results are simple and straightforward, we believe they are sufficient for justifying the calibration benefits of SAM.
>
> (**Q2**): The methodological contribution is not particularly innovative, as it mainly combines the existing SAM framework with focal loss. Thus, the degree of novelty—especially from a methodological perspective—may be limited in its ability to inspire readers.
>
> (**R2**): Thanks for the comment. While the modified loss function
> $$\tilde{\ell} _ {\tilde{\boldsymbol{\theta}}}(z) = \begin{cases}
> 		-\log \tilde{\mathbf{p}} _ {y}, &  \mathrm{if}\ \tilde{\mathbf{p}} _ {y}\leq 1/2, \\
> 		- \left(1 + \tilde{\mathbf{p}} _ {y}\right)^{-\gamma}\log \tilde{\mathbf{p}} _ {y}, & \mathrm{otherwise},
> 	\end{cases}$$
> 	is similar to the Focal Loss $\tilde{\ell} _ {\tilde{\boldsymbol{\theta}}}(z) = -(1 - \tilde{\mathbf{p}} _ {y})^\gamma \log \tilde{\mathbf{p}} _ {y}$.
> It should be highlighted that they are essentially quite different. Because Focal Loss is designed to deal with hard/minor examples, whereas our loss is designed to deal with over-confident examples. Another important difference is that our loss is particularly defined as the loss function of the SAM descent step. While we reduce the weight of over-confident examples, their contribution to the gradient does not vanish. In contrast, Focal Loss significantly suppresses the gradient contribution of the easy examples because the gradient of hard examples dominates the descent direction. This could nevertheless hurt the learning process and thus impede the generalization.
>
> To see this, we trained a ResNet-56 and a WRN-28-10 on CIFAR-100 with Focal Loss, Focal Loss + SAM (namely, both ascent step and descent step use Focal Loss as the loss function), and  Focal Loss + CSAM  (namely, only the ascent step uses Focal Loss as the loss function). As shown in the following table, we can observe that CSAM significantly outperforms SAM in terms of generalization. This suggests that prioritizing the hard examples is not always as effective as expected. Therefore, in contrast to Focal Loss, our proposed loss urges us to recognize the importance of easy examples during training, which might be critically important in the presence of data noise.
>
> || |  Focal Loss (FL) | SAM + FL| CSAM  + FL |
> |----|---|------|---|---|
> |ResNet-56 | Test Acc (%)	|  71.96 $\pm$ 0.28	| 73.11 $\pm$ 0.01 	| **73.81 $\pm$ 0.10** 	|
> | |  ECE (%)|   8.25 $\pm$ 0.23   	| 6.89 $\pm$ 0.38 	| **3.94 $\pm$ 0.17** 	|
> |WRN-28-10 | Test Acc (%)	|   80.55 $\pm$ 0.17    	| 81.33 $\pm$ 0.23 	| **82.08 $\pm$ 0.09** 	|
> | |  ECE (%) |   2.84 $\pm$ 0.36   	| 2.41 $\pm$ 0.06 	| **2.08 $\pm$ 0.11** 	|

---

> > ### Author Response · Authors · 2025-11-19
> > **Continued**
> >
> > (**Q3**):  Table S4 shows improved calibration results when 𝛾=0. Does this suggest that the proposed enhancement does not significantly improve calibration performance? At 𝛾=0, the method reduces to the standard SAM framework.
> >
> > (**R3**): Just as you said, at $\gamma=0$, the method reduces to the standard SAM framework. However, it should be noted that the improvement is not marginal.  Actually, the reason why the improvement seems to be marginal is that SAM has already attained SOTAs on many tasks, and the absolute value of ECE (%) is very small. In the following table, we visualize the improvement of ECE (data from Table S4) using relative change. So, as you may see, CSAM indeed significantly reduces the ECE value when compared to SAM.
> >
> > |           	|          	|             	| $\gamma=0.0$     	| $\gamma=0.5$     	| $\gamma=1.0$     	| $\gamma=1.5$     	| $\gamma=2.0$     	|
> > |--|--|--|-|--|--|--|--|
> > |      CIFAR-10       	| ECE      	| $\rho=0.05$ 	| 1.77 $\pm$ 0.04  	| 1.46 $\pm$ 0.14  	| 1.09 $\pm$ 0.07  	| 0.66 $\pm$ 0.06  	| **0.65 $\pm$ 0.14(-63.28%)**  	|
> > |           	|          	| $\rho=0.1$  	| 0.82 $\pm$ 0.12  	| **0.54 $\pm$ 0.11(-34.15%)**  	| 0.69 $\pm$ 0.14  	| 1.34 $\pm$ 0.33  	| 2.04 $\pm$ 0.17  	|
> > |     CIFAR-100       	| ECE      	| $\rho=0.05$ 	| 8.83 $\pm$ 0.15  	| 7.52$\pm$ 0.14   	| 6.24 $\pm$ 0.28  	| 4.86 $\pm$ 0.37  	| **3.67 $\pm$ 0.12(-58.43%)**  	|
> > |           	|          	| $\rho=0.1$  	| 6.17 $\pm$ 0.18  	| 4.34 $\pm$ 0.45  	| 3.26 $\pm$ 0.40  	| 1.84 $\pm$ 0.14  	| **1.42 $\pm$ 0.45(-76.98%)**  	|
> >
> > (**Q4**):  The conclusion states: “We proved that SAM achieves this goal by imposing an implicit regularization on the negative entropy of the predictive distribution during training, which is similar to focal loss.” Could the authors clarify which mathematical expression supports this statement? No corresponding equation appears in the theoretical derivations or references. A more detailed explanation would be appreciated.
> >
> > (**R4**): Sorry for the confusion. Actually, the reason why focal loss improves calibration is discussed in Section 4 of **Mukhoti et al., 2020** and the conclusion that SAM implicitly maximizes the predictive entropy is now given in Equation (1). We have explicitly added these details in the current version. Please see Lines 266 and 482.
> >
> > - Jishnu Mukhoti, Viveka Kulharia, Amartya Sanyal, Stuart Golodetz, Philip Torr, and Puneet Dokania. Calibrating deep neural networks using focal loss. In NeurIPS, pp. 15288–15299, 2020.
> >
> > (**Q5**):  Since the proposed method is derived from the theoretical analysis, can the assumptions established in the theory be satisfied in practical applications? In other words, do the conclusions derived under these assumptions hold in real-world optimization scenarios?
> >
> > (**R5**): Thanks for the comment. Actually, **we have trained a ResNet-56 on CIFAR-10 and a vision transformer (ViT) on ImageNet-1K to verify this assumption**. As shown in Figure 2(a) and Figure S4(a), we can observe that this assumption can be easily satisfied. So, generally,  the boundedness assumption is reasonable, and as a result, the conclusions hold even in real-world optimization scenarios.
> >
> > At last, many thanks for your time and effort. Please let us know if you have further questions. And if you are satisfied with our response, please kindly raise the score.

---

> > > ### Comment · Reviewer_AfCQ · 2025-11-28
> > > **Response to Rebuttal**
> > >
> > > Thank you for your detailed response. I acknowledge that the authors have indeed made a theoretical contribution. However, I still maintain some reservations regarding the limitations of the methodological innovations. I will consider adjusting my evaluation accordingly.

---

> > > > ### Author Response · Authors · 2025-11-28
> > > >
> > > > Dear Reviewer AfCQ,
> > > >
> > > > Many thanks for your positive feedback. While our proposed CSAM can further substantially improve the calibration performance of SAM (e.g., see  **R3** of <https://openreview.net/forum?id=c0ERcCz6lD&noteId=YeCit3CkmQ>), it should be noted that there is also a tradeoff between generalization and calibration for SAM. This is why we introduce an additional hyper-parameter $\gamma$. The experimental results suggest that, like Focal Loss, our proposed CSAM is widely applicable and useful as well. Of course, extending CSAM to other applications remains to be investigated.
> > > >
> > > > Furthermore, **the primary goal of our work is to reveal the calibration benefits of SAM and CSAM is just a plain extension inspired the theoretical analysis**. Actually, the effect of the training optimizer on the calibration performance has previously been little explored. For example, an interesting work, **Tao et al. 2024**, studies the calibration from many aspects, including post-hoc processing, bin size, architectural design, etc. But, unfortunately, they do not investigate the effect of optimizer as well. So, our work would encourage more researchers to revist the importance of training optimizer on calibration. From this point, we believe that our work would be highly valuable to the community.
> > > >
> > > > - Tao et al., A Benchmark study on calibration. ICLR, 2024.
> > > >
> > > > Thank you once again and please let us know if you have further questions.

---

### Official Review · Reviewer_nCsi · 2025-11-01

**Soundness:** 2
**Presentation:** 3
**Contribution:** 3
**Rating:** 8
**Confidence:** 4

**Summary:**

This paper investigates the calibration benefits of Sharpness-Aware Minimization (SAM) in deep neural networks. The authors provide a theoretical analysis showing that SAM implicitly regularizes the predictive distribution by encouraging higher entropy, thereby mitigating overconfidence. Specifically, they derive a lower bound and building on this insight they propose Calibrated SAM (CSAM), a variant that further enhances calibration by adaptively reweighting overconfident predictions. Extensive experiments on CIFAR, ImageNet-1K, and corrupted datasets demonstrate that both SAM and CSAM achieve significantly lower Expected Calibration Error (ECE) than standard training and existing calibration methods, without sacrificing accuracy.

**Strengths:**

(1) The paper’s key strength lies in its originality and significance: it provides the first formal theoretical explanation for why Sharpness-Aware Minimization (SAM) improves calibration—linking it to implicit entropy maximization—offering a principled understanding beyond empirical observation. This insight bridges optimization geometry and uncertainty quantification, a valuable contribution to both communities.

(2) The proposed CSAM variant is a high-quality and practical extension that consistently improves calibration across diverse architectures and datasets, including ImageNet-1K and distribution-shifted settings, without sacrificing accuracy.

(3) The paper is also clearly written, with well-structured theory, intuitive figures (e.g., reliability diagrams, Hessian analysis), and thorough experiments that effectively support its core message.

**Weaknesses:**

(1) Missing summation symbol in the ECE estimator (Lines 177–178). The definition of bin accuracy in the Expected Calibration Error (ECE) computation omits the summation over samples in bin. I think the correct expression should be  $\text{acc}(B_i) = \frac{1}{|B_i|} \sum_{z_j \in B_i} \mathbb{I}[y_j = \arg \max f_\theta(x_j)]$.

(2) The theoretical analysis (Lemma 1 and Theorem 1) relies on an assumption about the lower bound of the smallest Hessian eigenvalue along the linear interpolation between $\theta$ and $\theta'$. However, this assumption is only verified on a single model (ResNet-56) and a single dataset (CIFAR-10) in Figure 2. The generality of this assumption—and thus the applicability of the theoretical conclusions—remains unclear for other architectures (e.g., ViTs) or more complex datasets (e.g., ImageNet).

**Questions:**

(1) The assumption on the Hessian’s smallest eigenvalue (Lemma 1) is only validated for ResNet-56 on CIFAR-10 (Figure 2). Given that the theoretical conclusions rely on this assumption, do the authors have evidence—e.g., from preliminary experiments on ImageNet or ViTs—that this behavior (linear decay of  $k_{min}$ along $\theta$ to $\tilde{\theta}$) holds more broadly? If not, could they discuss potential failure modes where the assumption breaks down and how that might affect SAM’s calibration benefits?

(2) In the ECE estimator (Section 3.1, lines 177–178), the expression for $\text{acc}(B_i)$ appears to miss a summation over samples in the bin. Is this a typographical error? If yes, could it be corrected for formal precision?

---

> ### Author Response · Authors · 2025-11-19
>
> Many thanks for the valuable comments, and they are addressed one by one as follows.
>
>  (**Q1**): Missing summation symbol in the ECE estimator (Lines 177–178).
>
> (**R1**): Thanks for pointing out this typographical error. In line 177, we have corrected it as follows:
> $\operatorname{acc}(B _ i) = 1/|B _ i|\sum _ {z _ j\in B _ i} \mathbb{I}[y _ j=\operatorname{arg max}  f _ {\boldsymbol{\theta}}(x _ j)]$.
>
> (**Q2**): The theoretical analysis (Lemma 1 and Theorem 1) relies on an assumption about the lower bound of the smallest Hessian eigenvalue along the linear interpolation between $\boldsymbol{\theta}$ and $\boldsymbol{\tilde{\theta}}$. However, this assumption is only verified on a single model (ResNet-56) and a single dataset (CIFAR-10) in Figure 2. The generality of this assumption—and thus the applicability of the theoretical conclusions—remains unclear for other architectures (e.g., ViTs) or more complex datasets (e.g., ImageNet).
>
> (**R2**): Thanks for the comment. Actually, we further trained a vision transformer (ViT) on ImageNet-1K to verify this assumption. As shown in Figure S4(a), we can observe that this assumption can be easily satisfied as well. So, generally,  the boundedness assumption is reasonable, and as a result, the proof holds even for complex neural networks.
>
> (**Q3**):  The assumption on the Hessian’s smallest eigenvalue (Lemma 1) is only validated for ResNet-56 on CIFAR-10 (Figure 2). Given that the theoretical conclusions rely on this assumption, do the authors have evidence—e.g., from preliminary experiments on ImageNet or ViTs—that this behavior (linear decay $\kappa _ {min}$ of along $\boldsymbol{\theta}$ to $\boldsymbol{\tilde{\theta}}$ ) holds more broadly? If not, could they discuss potential failure modes where the assumption breaks down and how that might affect SAM’s calibration benefits?
>
> (**R3**): Following the last comment, we actually can observe the linear decay behavior of $\kappa_{min}$ even for ViT trained on ImageNet-1K (see Figure S4-b).
>
> (**Q4**):  In the ECE estimator (Section 3.1, lines 177–178), the expression for appears to miss a summation over samples in the bin. Is this a typographical error? If yes, could it be corrected for formal precision?
>
> (**R4**): Yes, this is a typographical error, and we have corrected it in Line 177.
>
> At last, many thanks for your time and effort. We hope that we have addressed all your concerns.

---

### Official Review · Reviewer_pqTz · 2025-11-01

**Soundness:** 3
**Presentation:** 3
**Contribution:** 2
**Rating:** 4
**Confidence:** 2

**Summary:**

This paper studies why Sharpness-Aware Minimization (SAM), beyond its known generalization benefits, tends to produce better-calibrated models. The authors show theoretically that SAM implicitly maximizes the entropy of the predictive distribution, discouraging overconfident outputs. Thus, they propose Calibrated SAM (CSAM), a simple variant that further suppresses overconfident examples through a lightweight modification to the loss. Experiments on CIFAR-10/100 and ImageNet-1K confirm that SAM already improves calibration compared to SGD or AdamW, and CSAM achieves the lowest Expected Calibration Error (ECE) without sacrificing accuracy.

**Strengths:**

## Strengths

* This paper provides a simple and practical variant. CSAM requires only a minor change and no additional computation, yet consistently improves ECE.
* The experiments are broad, covering both in- and out-of-distribution evaluations across convolutional and transformer-based models.
* The finding that SAM alone can outperform post-hoc calibration methods like temperature scaling is practically valuable for reliability-sensitive applications.
* The paper gives a clear mathematical explanation of why SAM tends to yield better-calibrated predictions, connecting it to implicit entropy regularization.

**Weaknesses:**

## Weaknesses

* The empirical effect has been reported before. The observation that SAM improves calibration is not new—Zheng et al. (2021) showed similar improvements in long-tailed recognition, and the original SAM paper (Foret et al., 2021) also mentioned better reliability qualitatively.
* The theoretical assumptions are strong. The derivations rely on smoothness and bounded-Hessian assumptions, which may not fully hold for complex networks, limiting the generality of the proofs.
* While CSAM consistently reduces calibration error, its advantage over SAM is small.

**References**

* Foret, P., Kleiner, A., Mobahi, H., & Neyshabur, B. Sharpness-Aware Minimization for Efficiently Improving Generalization. ICLR, 2021.
* Zheng, Z., Yang, X., Wang, Y., Li, Z., Liu, Y., & Zhang, T. Improving Calibration for Long-Tailed Recognition. CVPR, 2021.

**Questions:**

See weaknesses

---

> ### Author Response · Authors · 2025-11-19
>
> Many thanks for the valuable comments, and they are addressed one by one as follows.
>
>  (**Q1**): The empirical effect has been reported before. The observation that SAM improves calibration is not new—Zheng et al. (2021) showed similar improvements in long-tailed recognition, and the original SAM paper (Foret et al., 2021) also mentioned better reliability qualitatively.
>
> (**R1**): Just as you said, the observation that SAM improves calibration has been reported in Zheng et al. (2021) before, and we have claimed this fact in Line 077. However, **the reason why SAM can improve calibration remains under-explored**. In this paper, we attempt to formally answer this question and show that SAM implicitly maximizes the predictive entropy during training (Theorem 1) and consequently propose a SAM variant (namely, CSAM) to further reduce the calibration error. So, the core contribution of the paper is not to simply report an empirical observation, but instead to explain its underlying mechanism.
>
> -  Zheng, Z., Yang, X., Wang, Y., Li, Z., Liu, Y., & Zhang, T. Improving Calibration for Long-Tailed Recognition. CVPR, 2021.
>
> (**Q2**): The theoretical assumptions are strong. The derivations rely on smoothness and bounded-Hessian assumptions, which may not fully hold for complex networks, limiting the generality of the proofs.
>
> (**R2**): Thanks for the kind reminder. We understand that you are concerned that the assumptions are unrealistic. First, **we'd like to clarify that we do not require a smoothness assumption**. Second, the boundedness assumption only requires **the smallest eigenvalue of the Hessian not to be too small**, which is much looser than the smooth assumption. Actually, we further trained a ResNet-56 on CIFAR-10 and a vision transformer (ViT) on ImageNet-1K to verify this assumption. As shown in Figure 2(a) and Figure S4(a), we can observe that this assumption can be easily satisfied. So, generally,  the boundedness assumption is reasonable, and as a result, the proof holds even for complex neural networks.
>
> (**Q3**):  While CSAM consistently reduces calibration error, its advantage over SAM is small.
>
> (**R3**): Thanks for pointing out this issue. Actually, the reason why the improvement seems to be marginal is that SAM has already attained SOTAs on many tasks, and the absolute value of ECE (%) is very small. In the following table, we visualize the improvement of ECE (data from Table S7) using relative change. So, as you can see, CSAM indeed significantly reduces the ECE value when compared to SAM.
>
>
> |               	|                	| SAM             	| CSAM                      	|
> |---------------|---------------|----------------|-------------------------|
> | CIFAR-10      	| ResNet-56      	| 0.64 $\pm$ 0.09 	| 0.58 $\pm$ 0.07 (-9.37%)  	|
> |               	| WRN-28-10      	| 0.86 $\pm$ 0.13 	| 0.50 $\pm$ 0.03 (-41.86%) 	|
> |               	| PyramidNet-110 	| 0.74 $\pm$ 0.08 	| 0.32 $\pm$ 0.06 (-56.76%) 	|
> | CIFAR-100     	| ResNet-56      	| 1.66 $\pm$ 0.16 	| 0.84 $\pm$ 0.15 (-49.39%) 	|
> |               	| WRN-28-10      	| 2.11 $\pm$ 0.17 	| 1.50 $\pm$ 0.07 (-28.91%) 	|
> |               	| PyramidNet-110 	| 1.91 $\pm$ 0.14 	| 1.69 $\pm$ 0.04 (-11.52%) 	|
> | Tiny-ImageNet 	| ResNet-18      	| 3.46 $\pm$ 0.15 	| 2.75 $\pm$ 0.47 (-20.52%) 	|
>
> At last, many thanks for your time and effort. Please let us know if you have further questions. And if you are satisfied with our response, please kindly raise the score.

---

> > ### Comment · Reviewer_pqTz · 2025-11-26
> >
> > The authors’ rebuttal addressed most of my concerns, and I have raised my score. Thank you.

---

> > > ### Author Response · Authors · 2025-11-27
> > >
> > > Many thanks for the positive feedback. Please let us know if you have further questions.

---

### Comment · Area_Chair_6ZEx · 2025-11-25

Dear reviewers,

The authors have responded. We kindly ask you to review the authors' responses to your comments and provide your feedback. Thank you.

Best,

AC

---

### Meta-Review · Area_Chair_WGHJ · 2026-01-06

**Summary:**

This submission investigates the relationship between SAM—a widely used training mechanism—and calibration. Although prior work has observed that SAM improves calibration, the authors of this paper further analyze the phenomenon and provide a theoretical explanation for the underlying cause. Additionally, they propose an improved method that demonstrates greater robustness than SAM.

The initial scores for this submission showed significant divergence. The authors provided detailed responses during the rebuttal. It is noted that one reviewer with a score of 4 indicated she/he would raise their score, and another with a score of 2 stated they would increase it to 6. In light of this, I think the final average score for this submission should be positive. Given the current state of the reviews, the final decision for this paper should be accept. However, several critical points raised by the reviewers warrant further consideration by the authors:

(1) How does this explanation address deep models that are inherently underconfident?

**Reviewer Concerns:**

**Concerns addressed:**

*There is a prior work noted SAM's calibration improvement.*

Authors clarified their core contribution is the theoretical explanation (entropy maximization) and the proposal of CSAM.

*The theoretical framework of this paper is similar to that of [Calibrating Deep Neural Networks using Focal Loss], which operates by penalizing entropy to suppress overconfidence.*

The explanation on the non-trivialness of the connection between focal loss and SAM make sense.

**Still outstanding concerns:**

(1) The method proposed in this paper is relatively simple and lacks significant innovation.

(2) More comprehensive experimental validation would further strengthen the paper.

**Reviewer Scores:**

Reviewer pqTz has explicitly stated, "The authors’ rebuttal addressed most of my concerns, and I have raised my score." So the final score of reviewer pqTz could be changed to 6.

Reviewer 7xpcI stated that she/he will increase her/his socre to 6.

Reviewer AfCQ acknowledged the theoretical contribution of this submission but maintained reservations on methodological innovation, stating they would "consider adjusting her/his evaluation accordingly."

---

### Decision · Program_Chairs · 2026-01-26

Accept (Poster)